# Ultra-fast deep-learned CNS tumour classification during surgery

C. Vermeulen[1,2,6], M. Pagès-Gallego[1,2,6], L. Kester[3], M. E. G. Kranendonk[3], P. Wesseling[3,4], N. Verburg[5], P. de Witt Hamer[5], E. J. Kooi[4], L. Dankmeijer[4,5], J. van der Lugt[3], K. van Baarsen[3], E. W. Hoving[3], B. B. J. Tops[3 ✉] & J. de Ridder[1,2 ✉]

Central nervous system tumours represent one of the most lethal cancer types, particularly among children[1]. Primary treatment includes neurosurgical resection of the tumour, in which a delicate balance must be struck between maximizing the extent of resection and minimizing risk of neurological damage and comorbidity[2,3]. However, surgeons have limited knowledge of the precise tumour type prior to surgery. Current standard practice relies on preoperative imaging and intraoperative histological analysis, but these are not always conclusive and occasionally wrong. Using rapid nanopore sequencing, a sparse methylation profile can be obtained during surgery[4]. Here we developed Sturgeon, a patient-agnostic transfer-learned neural network, to enable molecular subclassification of central nervous system tumours based on such sparse profiles. Sturgeon delivered an accurate diagnosis within 40 minutes after starting sequencing in 45 out of 50 retrospectively sequenced samples (abstaining from diagnosis of the other 5 samples). Furthermore, we demonstrated its applicability in real time during 25 surgeries, achieving a diagnostic turnaround time of less than 90 min. Of these, 18 (72%) diagnoses were correct and 7 did not reach the required confidence threshold. We conclude that machine-learned diagnosis based on low-cost intraoperative sequencing can assist neurosurgical decision-making, potentially preventing neurological comorbidity and avoiding additional surgeries.

The most common first line treatment for central nervous system (CNS) tumours is neurosurgical resection of the tumour. An important factor for determining whether the risk of a more aggressive resection is acceptable is the tumour type. For instance, diffuse midline gliomas with a specific histone H3 (H3K27) mutation are considered incurable, indicating that surgery should primarily be aimed at acquisition of tumour tissue for diagnosis and preserving quality of life, rather than attempting complete resection[5]. Similarly, medulloblastomas show limited prognostic improvement between near-total and total resection, indicating that maximal resection is not necessarily preferable for these tumours[6]. However, radical resection is beneficial for other tumour types: in posterior fossa ependymoma type A and atypical teratoid rhabdoid tumour, a strategy of aiming for gross total resection should be followed, since this is an important prognostic factor[7–10]. Moreover, in CNS tumours in adults, the extent of resection matters: gross total resection has been reported to offer survival benefits for isocitrate dehydrogenase (IDH)-wild-type glioblastoma of the receptor tyrosine kinase (RTK) I and RTK II subtypes, but not for the mesenchymal subtype[11]. Similarly, in IDH-mutant astrocytoma, overall survival is negatively affected when gross total resection is not achieved[12]. The neurosurgical strategy thus depends on a precise and reliable diagnosis of the tumour.

Altered genome-wide DNA methylation patterns are highly distinctive features of neoplasms, and the assessment of DNA methylation can reveal information about the origin and prognosis of a tumour[13–16]. High-dimensional CpG methylation profiles can be accurately assigned to a specific CNS subtype using machine learning approaches, in particular random forest classification[14,15]. Methylation arrays[17,18], in combination with the algorithm described by Capper et al.[14], are widely used in routine diagnostic practice. However, the turnaround time for obtaining array-based methylation profiles is in the order of several days and therefore incompatible with an intraoperative setting.

Current practice consists of preoperative imaging and intraoperative diagnosis achieved by rapid histological assessment of frozen tumour sections. However, this does not always result in a clear diagnosis, and the provisional frozen-section diagnosis is sometimes revised on the basis of post-operative tissue-based diagnostics. As a result, some patients require a second surgery, whereas others could in hindsight have been operated less radically.

Nanopore DNA sequencing has recently emerged as a method that enables ultrarapid sequencing-based diagnosis[19,20]. Major advantages of nanopore sequencing include its low setup cost, small form factor and instant data availability. In addition, nanopore sequencing enables

[1]Oncode Institute, Utrecht, The Netherlands. [2]Center for Molecular Medicine, UMC Utrecht, Utrecht, The Netherlands. [3]Princess Máxima Center for Pediatric Oncology, Utrecht, The Netherlands. [4]Department of Pathology, Amsterdam University Medical Centers/VUmc, Amsterdam, The Netherlands. [5]Department of Neurosurgery, Amsterdam University Medical Centers/VUmc, Amsterdam, The Netherlands. [6]These authors contributed equally: C. Vermeulen, M. Pagès-Gallego. ✉e-mail: b.b.j.tops@prinsesmaximacentrum.nl; j.deridder-4@umcutrecht.nl

direct measurement of methylated cytosines and a substantial reduction in sample preparation times[21]. Thus a tissue sample can be sent for sequencing in the early stages of surgery to obtain a molecular diagnosis in time to affect and shape the neurosurgical strategy[4]. A major challenge of this application is that only very sparse methylation profiles can be generated in such a short time. Moreover, it is a priori unknown which CpG sites will be covered.

To enable tumour classification in an intraoperative setting, we have developed Sturgeon, a neural network classifier that is patient-agnostic and optimally tuned to deal with sparse data. In the Sturgeon approach, extensive computational resources are allocated to train and validate complex neural networks prior to surgery. This is a major advantage over existing classification algorithms that rely on patient-specific model training during surgery[4]. Our final models are trained on 36.8 million simulated nanopore runs and validated on a further 4.2 million simulated nanopore runs (Fig. 1). This enables us to extensively validate and calibrate our models prior to applying them. The resulting Sturgeon model is portable and takes only a few seconds to run on a laptop computer.

As a proof of concept, we trained Sturgeon models for CNS tumour classification and retrospectively applied them on sparse nanopore sequencing data in 50 CNS tumour samples and 415 publicly available[22] nanopore-sequenced CNS samples. The model was able to correctly classify the vast majority of samples (45 out of 50) based on data equivalent to 20–40 min of sequencing, in line with a 90-min time window between obtaining tissue in the operating room and diagnosis. Finally, we demonstrate the ability of Sturgeon to influence surgical decision-making by applying it in a realistic intraoperative setting for 25 CNS tumour resections.

## Neural network training via simulation

Within a turnaround time of 60–90 min, only very limited nanopore sequencing data—on the order of 100–400 Mb—can be generated. As a result, extremely sparse coverage across the genome is expected (covering 0.5–4% of the CpG sites in a 450K array). As it is a priori unknown which sites will be covered, this poses a substantial challenge for the downstream machine learning model. There is a lack of large, well-annotated nanopore-based methylation datasets and it will take years to reach the comprehensiveness of the available array-based datasets. We therefore developed a simulation strategy that generates realistic training data from array-based methylation profiles. Finally, effectively training neural network models requires orders of magnitude more training samples than the number of patient samples available. Sturgeon therefore uses a data-augmentation approach to upsample the number of training samples available: thousands of unique shallow nanopore sequencing experiments are simulated from each methylation profile.

Sturgeon is designed to train a neural network on simulated nanopore sequencing runs. Here, we used the publicly available Infinium 450 K profiles reported in Capper et al.[14]. This dataset contains 2,801 reference labelled methylation profiles from CNS tumour and control tissue samples. The simulation consists of the following components (Fig. 1a): (1) binarization of the array beta values to either methylated or unmethylated state, to account for the limitation that in shallow sequencing the expected coverage is ≤1× for the vast majority of detected sites, precluding the ability to reflect heterogeneously methylated sites; (2) non-uniform CpG site sampling to account for the fact that nanopore sequence reads are approximately 5 kb in length (assuming the rapid sample preparation methods used in an intraoperative setting); (3) variable sampling of the number of CpG sites covered, to account for read accumulation as time progresses; (4) random noise, to account for the expected discrepancy rate of 10–15% in nanopore methylation-aware sequencing compared with binarized methylation arrays, resulting from a combination of heterogeneous methylation states across alleles

and cells, and due to methylation calling errors[23] (Supplementary Fig. 1 and Supplementary Table 1).

The simulated nanopore sequencing data are used to train four neural networks (Extended Data Fig. 1 and Methods). These neural networks are each trained, validated and calibrated independently (Fig. 1b). To this end, we split the Capper et al. reference dataset[14] into four folds while keeping the original class distributions. We then use two folds to train the submodel, one fold to determine the best-performing state of the submodel and to perform score calibration and the final fold to evaluate the submodel's performance.

We evaluated the performance of the Sturgeon submodels on the hold-out test set. The submodels achieved a F1 score of 0.935 across all classes at the approximate equivalent of 40 min of sequencing (Fig. 1c). Specific classes show an increased error rate, as may be expected owing to their biological similarity (for example, melanotic schwannoma versus schwannoma, different TSH (thyroid-stimulating hormone)-secreting pituitary adenoma subtypes and highly similar glioblastoma subclasses). When aggregating scores on the family level, performance is even higher, with an average F1 score of 0.984 at the same sequencing time (Extended Data Fig. 2b). As expected, Sturgeon's performance is directly correlated to the sequencing depth, and performance increases most markedly within the first 50 min of simulated sequencing with 0.6% to 4% of the 450K CpG sites covered (Extended Data Fig. 2c and Supplementary Figs. 2 and 3). We then calibrated the classifier scores to ensure that for example, a classifier score of 0.9 corresponds to the classifier being correct 90% of the time. For this purpose we applied temperature scaling[24]. As a result of temperature scaling, the overall expected calibration error (ECE), decreased from 0.025 to 0.002 in the test set (Supplementary Table 2 and Supplementary Figs. 4–7). We decided to conservatively use a cut-off score of 0.95 to confidently classify a sample. Using this cut-off, 80 out of the 91 classes in the test set have a true positive rate (TPR) higher than 0.95; With a less conservative threshold of 0.8, 26 out of 91 classes do not reach the expected 0.8 TPR (Extended Data Fig. 2d and Supplementary Fig. 8).

## Classification of paediatric array data

The training dataset for Sturgeon consists of a varied population of patients of different ages (mean age of 29 with 36% less than 13 years of age). We first aimed to further validate the performance of Sturgeon in a paediatric setting. For this purpose, we obtained 94 genome-wide methylation profiles generated using the Illumina Infinium MethylationEPIC v1.0 (hereafter referred to as EPIC) arrays from patients that underwent a CNS tumour resection surgery in the Princess Máxima Center (PMC) for paediatric oncology. For each of these samples, the publicly available Heidelberg classifier (v11b4) was applied as part of the routine clinical care. This classifier can be considered an updated version of the Capper et al. classifier[14]. The recommended cut-off for a clinical diagnosis[25] is 0.84, which the classifier reached for the majority ($n = 68$) of samples. Those classified below the 0.84 cut-off ($n = 26$) are considered difficult to diagnose based on their methylation profile, which is likely to occur for uncommon tumour types that do not correspond to any of the previously annotated classes, tumours that occur in the context of a genetic tumour predisposition syndrome, heterogeneous samples or samples with a low tumour purity.

For each methylation profile we simulated 500 nanopore sequencing experiments at 7 sequencing depths (Fig. 2a,b and Methods) for a total of 332,500 simulated nanopore sequencing experiments, after which we applied the Sturgeon classifier (Fig. 2, Extended Data Fig. 3 and Supplementary Table 3).

For cases with a clear diagnosis (Heidelberg score >0.84), Sturgeon classified correctly (at the 0.8 threshold) in 95.3% (32,412 out of 34,000 simulated samples) in as little as 25 min of simulated sequencing. At the conservative threshold of 0.95, 86.2% (29,316 out of 34,000) of simulated samples were correctly classified (Fig. 2a,

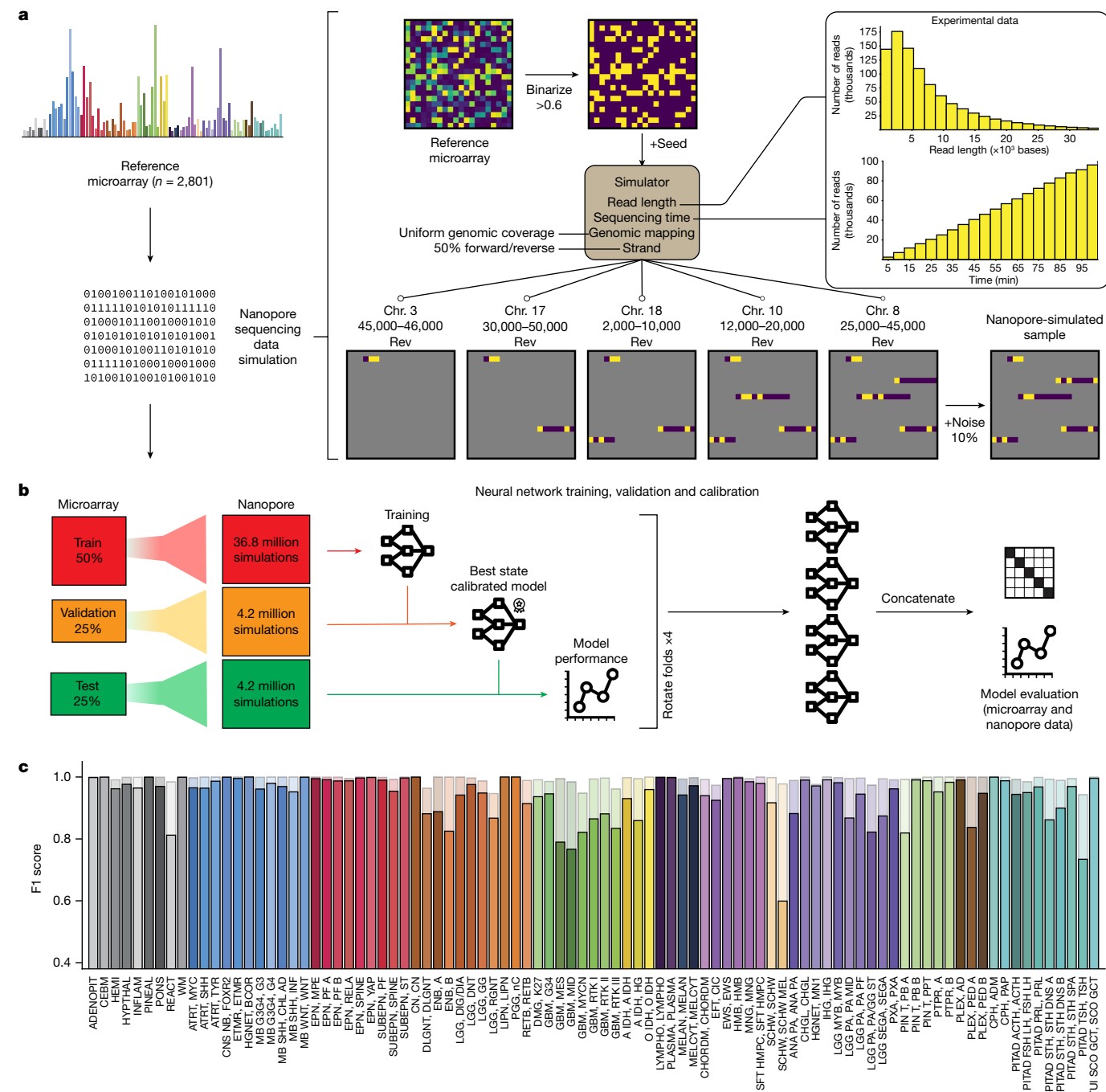

**Fig. 1 | Schematic representation of the simulation, cross-validation approach and results on simulated data. a**, Nanopore sequencing runs were simulated from the Capper et al. reference dataset[14] comprising 2,801 labelled methylation profiles from CNS tumour and control samples. Sequencing data were simulated on the basis of existing nanopore sequencing runs (read length distribution and throughput); as these simulations produce very sparse samples, millions of unique samples can be simulated. Chr. chromosome. **b**, Fourfold cross-validation was performed by rotating the folds to obtain four models that were used in the final prediction of external microarray data and nanopore sequencing data. **c**, Performance of Sturgeon on the four test folds of the Capper et al. dataset (added up for the four submodels). F1 scores for each reference label at 40 min of simulated sequencing (approximately 97% missing values compared with microarray data). Solid bars indicate the F1 score for the highest-scoring class and transparent bars show the F1 score for the top three highest-scoring classes. Complete class names, adapted from Capper et al.[14], can be found in Supplementary Table 2.

second timepoint). At the same timepoint, only 2.7% and 13.8% of simulations did not reach a confidence score exceeding 0.8 and 0.95 respectively. Incorrect diagnoses were called in 2.0% of simulations at the 0.8 threshold, and only 0.5% for the conservative 0.95 threshold. At 50 min of simulated sequencing (Fig. 2a, fourth timepoint) performance improved slightly, with 97.1% (33,020 out of 34,000) of simulations reaching a correct diagnosis with confidence ≥0.8 and

90.8% with a score ≥0.95. A total of 1.6% of simulations did not reach a score ≥0.8. Wrong diagnoses were called in only 1.3% of simulations with a score ≥0.8 and 0.5% with a score ≥0.95. Together, these results suggest that a conclusive diagnosis can be reached within 25–50 min of simulated sequencing for the vast majority of paediatric cases that can be classified using the Heidelberg v11b4 classifier, with a very low error rate.

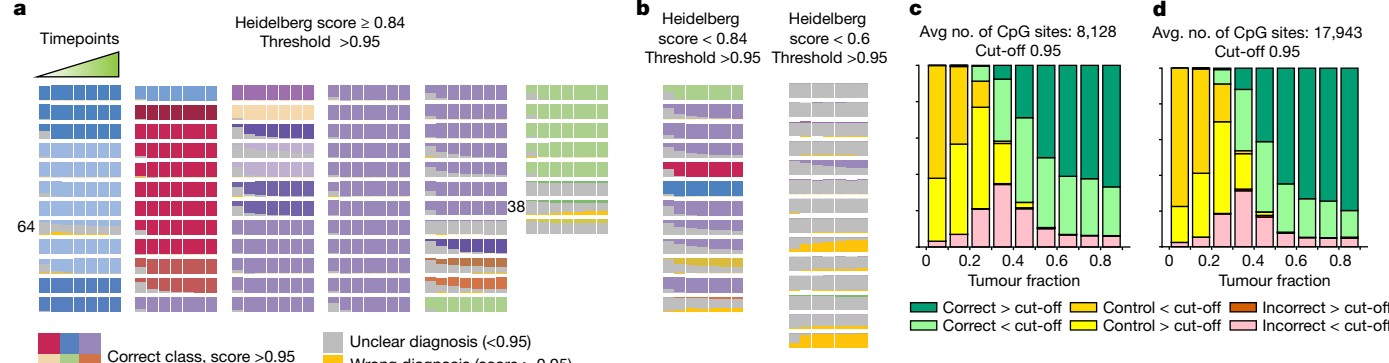

**Fig. 2 | Classification performance over time on nanopore runs simulated from paediatric CNS tumour methylation arrays. a–d,** For each of 94 methylation profiles, 500 experiments were simulated per timepoint, corresponding to approximately 12.5 min of additional sequencing per consecutive timepoint (Methods). **a,** Each series of bars corresponds to one of the 68 cases for which a clear Heidelberg classifier result was obtained (Heidelberg score >0.84). Bars indicate at each timepoint the proportion of outcomes when a 0.95 confidence score is used. The correct fraction is coloured by class; colours correspond to those in Fig. 1c. Unclear (no class reached a confidence score ≥0.95 or a control class reached a confidence score ≥0.95) classifications are shown in grey, wrong classes with a confidence score ≥0.95 are yellow. **b,** As **a,** but for the 26 samples for which the Heidelberg

classifier was inconclusive (Heidelberg score <0.84; unclear cases). **c,** Stacked bar graph to show the effect of different tumour fractions on classifier performance on short sequencing simulations (8,128 CpG sites covered on average, equivalent to roughly 20 min of sequencing). Nanopore sequencing experiments were simulated from the reference samples from the Capper et al. dataset[14]. The reported sample purity was used as a baseline and control tissue reads were added to simulate lower sample purities. The fraction of correct and incorrect classifications over or under the confidence threshold (≥0.95) are shown, as well as the number of simulations where the classifier predicted the sample as control tissue. Avg., average. **d,** same as **c,** with a higher sequencing depth (17,943 CpG sites covered on average, equivalent to approximately 40 min of sequencing).

The majority of misclassifications (141 out of 155 at timepoint 3) occurred in the simulated sequencing experiments from 2 samples (Fig. 2a). In both cases, the misclassification occurred within the same family (PMC_38: glioblastoma subtype midline misclassified as subtype H3K27 mutant; and PMC_64: medulloblastoma subtype group 4 classified as subtype group 3).

For the difficult-to-diagnose cases, Sturgeon generally performed less well. For most of these cases, a definitive diagnosis was reached based on the combination of molecular and histological features. In 11 of the 26 cases Sturgeon reached a diagnosis in concordance with the definitive diagnosis (but often at later timepoints) in the majority of simulations. All of these 11 cases also reached a Heidelberg classifier score between 0.6 and 0.84 (Fig. 2b and Supplementary Table 3). In the remaining cases, both Sturgeon and the Heidelberg classifier performed poorly, most frequently resulting in an unclear diagnosis (low confidence scores or high confidence scores for control tissue classes). This can be attributed to different reasons: low tumour fraction based on histology (PMC_1, PMC_28, PMC_82 and PMC_76); classes not present in the 2018 classification scheme (PMC_71, PMC_73, PMC_77 and PMC_88); no definitive diagnosis (PMC_72 and PMC_75); or tumours in the context of a germline mutation (PMC_89, PMC_85, PMC_91 and PMC_77), which has been suggested to complicate methylation-based classification[15] (see Supplementary Table 3 for further notes and Supplementary Figs. 9 and 10 for more details).

Together, these simulation results indicate that Sturgeon can perform on par with the Heidelberg v11b4 classifier, even when applied to a very sparse simulated sequencing run. It also reiterates the limitation that Sturgeon (as any other machine learning-based classifier) is only able to perform well in samples that are sufficiently represented in the training data. Reassuringly, for classes that are not represented in the training data, confidence scores are usually low, resulting in an unclear outcome rather than a misdiagnosis.

### Sample purity affects sensitivity

In an intraoperative setting, time constraints do not always allow for sample selection on the basis of purity and samples may therefore contain a larger fraction of normal cells. By contrast, the training dataset

consists of samples that are relatively pure, with the tumour cell content ranging from 40% to 85% (Extended Data Fig. 4a). We therefore aimed to further explore the behaviour of Sturgeon on samples with low tumour purity by in silico mixing of simulated nanopore reads from one of the control tissues included in the Capper et al. dataset[14] with simulated nanopore reads from a non-control sample (Extended Data Fig. 4b and Methods). Simulations show that as expected, a higher fraction of admixed control reads reduces performance, increasing the number of cases for which the classifier does not reach a confident classification. At lower (<50%) tumour fractions, the number of cases for which the control class is predicted increases (Fig. 2c,d and Extended Data Fig. 4c). Notably, admixing control tissue reads does not lead to significant numbers of misclassifications, indicating that high scores are reliable, even when the tumour fraction is unknown. Deeper sequencing does not seem to resolve the difficulties in classifying samples with low tumour fractions (Fig. 2d). To estimate how frequently this lower limit would not be reached in clinical practice, we retrospectively collected metadata from 44 cases in which intraoperative histology was performed. In 6 out of 44 paediatric cases the pathologist estimated the tumour fraction to be below 50%, in 5 out of 44 cases the pathologist estimated the tumour fraction to be around 50% and in 31 out of 44 cases the tumour fraction was estimated to be above this threshold (Supplementary Table 4). In two cases, the tumour cell fraction was not estimated. For adult glioma cases the mean tumour purity was estimated to be 69% in samples obtained from the enhancing fluid-attenuated inversion recovery region[26], indicating that in these types of samples, material of adequate purity can be obtained more consistently. On the basis of these results, we expect that in the intraoperative setting, especially in paediatric cases, some samples may not be classifiable owing to low tumour purity. We do not expect low tumour purity to result in misdiagnosis in these samples when using the most stringent cut-off.

### Classification of nanopore-sequenced samples

Next, we assessed the performance of Sturgeon on real nanopore sequencing data. We retrospectively sequenced and classified 27 paediatric brain tumour DNA samples obtained from the PMC biobank. We applied Sturgeon to increasing numbers of reads, mimicking an

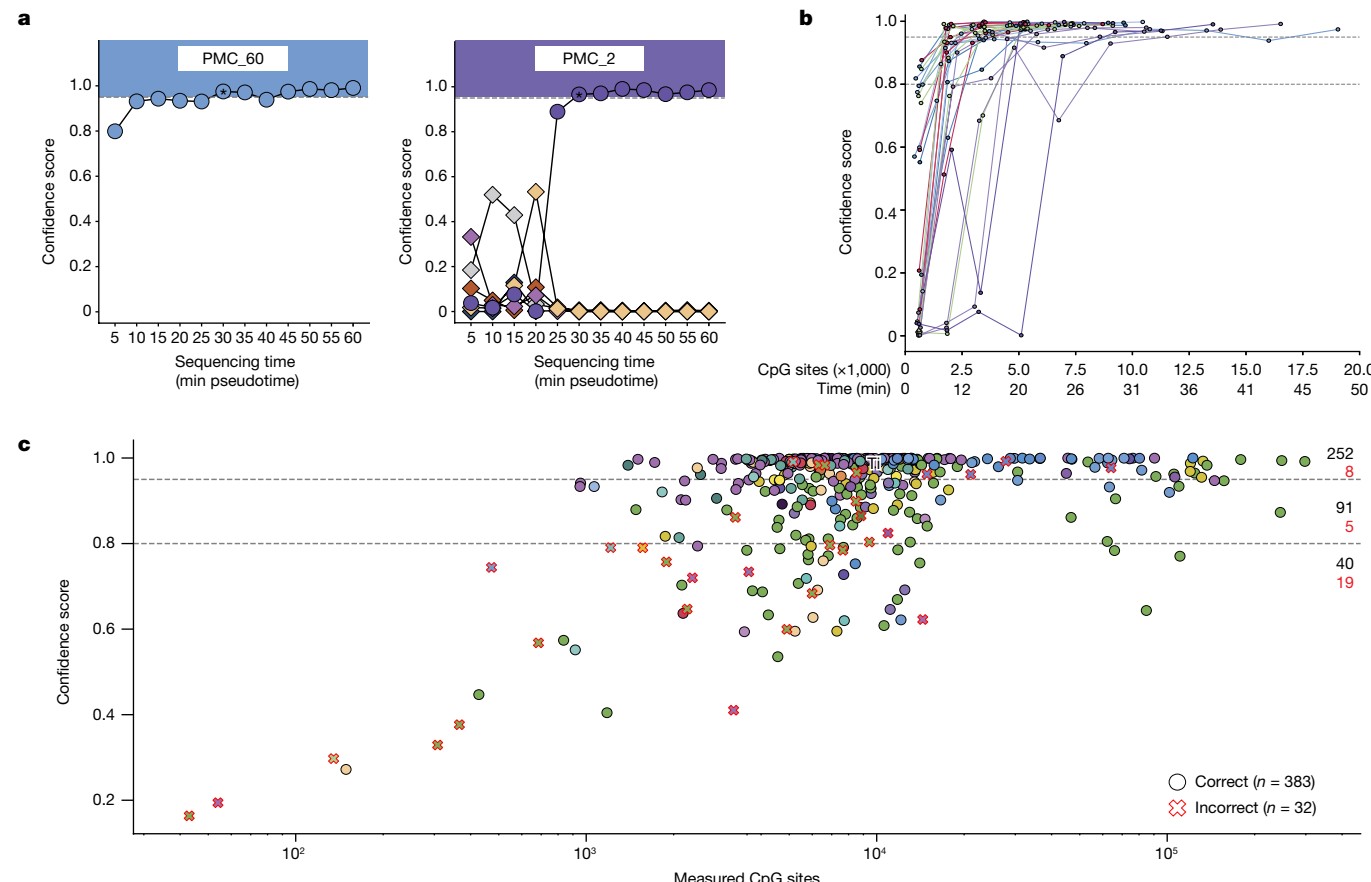

**Fig. 3 | Sturgeon applied to nanopore-sequenced samples. a**, Two representative examples of Sturgeon classification on nanopore-sequenced samples. The *x* axis indicates the sequencing time (5-min pseudotime intervals) and the *y* axis indicates the confidence score. Circles indicate the confidence score of the correct class; diamonds indicate the confidence score of incorrect classes (classes with averaged scores lower than 0.1 are omitted). Asterisks indicate the first timepoint at which the confidence score of the correct class was higher than 0.95. **b**, Sturgeon classification scores for 27 paediatric CNS

tumour samples at increasing sequencing time (5-min pseudotime intervals). Only the confidence score of the correct class is plotted; see Extended Data Fig. 5 for complete results for each sample. **c**, Sturgeon classification results on the publicly available data from Kuschel et al.[22] (415 sequencing runs). Samples for which the highest-scoring class is correct are indicated as circles and samples for which the highest-scoring class is incorrect are shown as crosses. Points are filled on the basis of the correct class according to the colour scheme shown in Fig. 1c.

average minION sequencing experiment in 5-min pseudotime intervals (Fig. 3a, Extended Data Fig. 5 and Methods).

The classification results demonstrate that for 24 out of 27 samples, Sturgeon assigned a score higher than 0.95 to the correct class after the equivalent of 25 min of sequencing; on average this threshold was achieved in 15–20 min of sequencing (Fig. 3a and Supplementary Table 5). Samples PMC_2, PMC_60 and PMC_29 reached the 0.95 threshold at 30, 30 and 35 min of sequencing, respectively. For the majority of samples, we generated around 200,000 reads (approximately 3.9 Gb), where a typical intraoperative run (60 min of sequencing) would be expected to yield approximately 60,000 reads (around 200 Mb) of throughput. This enabled us to evaluate the robustness of the results by randomly subsampling sequence reads, essentially simulating a different order in which the DNA molecules were sequenced. Our results show that Sturgeon is very robust, reporting the correct class in 27,980 (score ≥0.95) out of the 36,000 predictions (77%); and only confidently reporting the incorrect class 14 times (0.03%) (Supplementary Table 6). The outcomes are more confident and accurate if more sequence reads are available (Extended Data Fig. 6). This also showcases how some samples (for example PMC_2 and PMC_29) are more difficult to classify than others.

To further assess the robustness of Sturgeon and its susceptibility to batch and operator biases, we validated it on a publicly available dataset[22]

(GSE209865) consisting of nanopore sequencing data for 415 CNS tumour sequencing runs, corresponding to 382 unique samples (Fig. 3c, Supplementary Fig. 11 and Supplementary Table 7), including from adult patients. The provided dataset is already processed—where methylation calling is performed using Nanopolish (rather than Megalodon or Guppy, which we use in our default workflow)—and probe methylation status is already mapped. We find that despite these differences in sample workflow, Sturgeon still performs as expected, even slightly outperforming nanoDx, the patient-specific random forest classifier used in ref. 22, as it is able to correctly predict nine additional samples (Supplementary Fig. 11b). Sturgeon correctly classified 383 (92.2%) samples, 343 (82.6%) at a confidence threshold ≥0.8 and 252 (60.7%) samples with a confidence ≥0.95. From the 415 samples, 32 (7.7%) were incorrectly classified, 8 (1.9%) of which reached a confidence score ≥0.95 (Fig. 3c and Supplementary Fig. 11a). We note that, for five of these eight confidently incorrectly classified samples, four corresponded to a single sample that was also incorrectly classified by nanoDx, and one is incorrectly classified as a 'somatotropin hormone-producing pituitary adenoma' instead of a 'TSH-producing pituitary adenoma'. Overall, nanoDx is able to perform better in scenarios with extremely low coverage of CpG sites, owing to its patient-tailored model. However, Sturgeon performs better and is more confident with a CpG site coverage compatible with intraoperative sequencing (Supplementary Fig. 11).

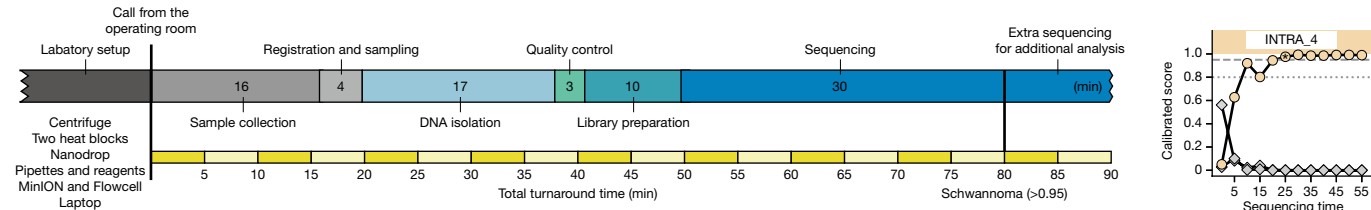

**Fig. 4 | Intraoperative sequencing turnaround time.** Timeline for surgery and intraoperative sample analysis for INTRA_4 (Supplementary Video https://zenodo.org/record/8261128), with the turnaround time and required time per processing step indicated (in minutes). Circles indicate the confidence score of the correct class; diamonds indicate the confidence score of incorrect classes (classes with averaged confidence scores lower than 0.1 are omitted). The asterisk indicates the first timepoint at which the score of the correct class was higher than 0.95.

## Copy number variations

In addition to methylation profiling, copy number variations (CNVs) have an important role in tumour classification, prognosis and downstream treatment. We explored whether CNVs could be detected using shallow nanopore sequencing to further support the classifications provided by Sturgeon. For example differentiating between IDH-mutant oligodendroglioma and astrocytoma can be challenging (Supplementary Fig. 13), and in such cases a chromosome 1p/19q codeletion is clear evidence of the former. To this end, we adapted the approach described by Euskirchen et al.[27]. Using a downsampling approach, we were able to detect large scale CNVs such as the chromosome 1p deletion from as few as 20,000–50,000 sequence reads (Extended Data Fig. 7 and Methods), although smaller CNVs such as the chromosome 19q deletion are less reliably detected. The clinical relevance of different CNVs is context-dependent and we therefore follow the example of the Heidelberg classifier: we provide CNV plots in our workflow to the pathologist in parallel with the methylation classifier result for further interpretation. More examples of CNV profiles derived from nanopore sequencing are shown in Supplementary Figs. 14 and 15.

## Intraoperative sequencing

To demonstrate the clinical feasibility of Sturgeon in an intraoperative sequencing context, we performed the protocol during 25 surgeries at two different hospitals in the Netherlands. This produced 20 paediatric samples from surgeries performed at the PMC and 5 adult samples from surgeries performed at the Amsterdam University Medical Centers (AUMC). Samples obtained for histological assessment during surgery were split, and one part was used for intraoperative sequencing and the other part was used for histological assessment. To rapidly obtain a high concentration of input DNA, we optimized the DNA extraction protocol to extract DNA from fresh tissue samples in 17–20 min (Methods).

Extended Data Table 1 lists the results and context of the 25 intraoperative sequencing experiments (Extended Data Fig. 8 and Supplementary Table 8 show the scores as they developed over time). The processing and analysis for sample PMC_live_4 was captured on film (Supplementary Video available for download from https://zenodo.org/record/8261128; the timeline for this particular sample is shown in Fig. 4).

For the five CNS tumour samples from adult patients, we focused specifically on glioma, as surgical strategy may affect the outcome differently in IDH-wild-type versus IDH-mutant high-grade gliomas[28,29]. In cases with a high-grade glioma, showing enhancement on T1 contrast-weighted MRI, intraoperative fluorescence (5-aminolevulinic acid) is used to mark tumour cells during resection[30], allowing consistent sampling of high tumour content samples and resulting in successful classification in 4 out of 5 cases where tumour was sampled.

In summary, Sturgeon was able to correctly classify 72% of tumours (18 out of 25) at the subclass level with at most 45 min of sequencing. In the other cases the classification was unclear, which can be attributed to tumour classes not present in the reference dataset (INTRA_11), low tumour purity (INTRA_1, INTRA_3, INTRA_14 and INTRA_15) or exotic cases (INTRA_8 and INTRA_13). We encountered several cases where a rapid molecular classification would have been of substantial added value. For example, in cases INTRA_23 and INTRA_25, the tumour class was not known prior to surgery. Intraoperative frozen-section diagnosis suggested an ependymoma in both cases, indicating a radical resection as the best course of action. Shortly afterwards, Sturgeon achieved a high confidence for ependymoma subtype *RELA* fusion and ependymoma subtype A, respectively. Both classifications were later confirmed using a methylation array and the Heidelberg V11b4 classifier. Obtaining the molecular class intraoperatively in these cases provided an independent classification that corroborated the intraoperative frozen-section diagnosis, reducing the risk of a misdiagnosis and providing additional certainty for the neurosurgeon to follow the best surgical plan for these patients.

## Location-specific models

The Capper et al. dataset[14] encompasses 81 tumour classes. However, many class distinctions are only relevant within a particular topological context, and could therefore already be ruled out prior to surgery. For instance, for a surgery of the spinal mass, a classifier generally does not need to be able to detect pituitary adenomas. Compared with other regions in the brain, the number of relevant classes in the brainstem is relatively low (*n* = 21; Supplementary Table 9). We hypothesized that, by merging the irrelevant classes from the training dataset into a single class, the model can focus on the truly relevant classes and improve its performance. To test this hypothesis, we developed a Sturgeon classifier specifically for brainstem tumours. In summary, this adjusted approach can slightly improve the required sequencing depth. The design, validation and results are discussed in Supplementary Note and Supplementary Figs. 16–21.

## Adaptive sampling

With the most recent chemistry, nanopore sequencers are able to reverse the current in specific channels, thereby ejecting reads as they are being sequenced. This has enabled 'adaptive sampling', in which a read is mapped when the first approximately 400 bases are sequenced and subsequently rejected if it falls outside any of the targeted regions[31]. This strategy can improve the turnaround time by rejecting reads that are unlikely to overlap informative CpG sites. We designed an adaptive sampling strategy (Methods) and tested this in five samples. Overall, the number of informative CpG sites sequenced per minute was 15–30% higher with adaptive sampling, and Sturgeon was more confident in classifying the same sample (Fig. 5a,b and Supplementary Figs. 22–24). However, we note that adaptive sampling can be technically challenging to set up on custom hardware and comes with an increased hardware requirement. For simplicity, we opted to not use it in an intraoperative setting.

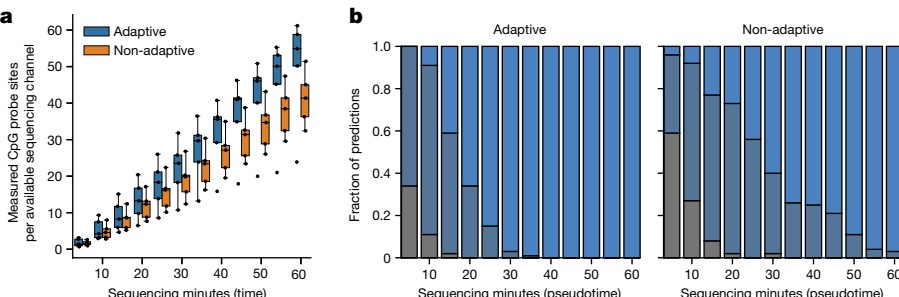

**Fig. 5 | Adaptive sampling can reduce turnaround time. a**, Five samples were run with adaptive sampling on half of the available channels. Box plots indicate the number of 450K array CpG probe sites during sequencing, normalized to the amount of available sequencing channels; minimum and maximum bounds represent the 25th and 75th percentiles, respectively; and the center bound represents the median; whiskers extend to 1.5 times the interquartile range. Dots indicate the underlying data. **b**, Robustness analysis results of sample PMC 68 on adaptive (left) and non-adaptive (right) channels (results for all samples are presented in Supplementary Fig. 24). Reads were accumulated in resampled orders ($n = 100$) at a rate based on the average MinION sequencing speed. Sturgeon was then applied to all permutations. Colours indicate the type of prediction made by Sturgeon: plain colour (correct class and score ≥0.95), darkened colour (correct class and score <0.95) and grey (incorrect class and score <0.95).

## Discussion

Here we demonstrate the practical feasibility of intraoperative methylation-aware nanopore sequencing for paediatric and adult CNS tumour classification that can be used to improve surgical decision-making. Classification during surgery is challenging because owing to the short sequencing time, only very sparse data are available and it is a priori unknown which CpG sites will be covered. Furthermore, nanopore-sequenced reference samples are not widely available.

To address these challenges, we developed Sturgeon, a deep learning approach that is trained on simulated nanopore sequencing data generated from readily available methylation array data and can accurately classify tumour types based on intraoperatively generated sequence data. Sturgeon uniquely moves the computationally intensive model training, validation and calibration phase outside the surgical time window, providing well-tested highly accurate one-size-fits-all classification models. In contrast to previous approaches[4,22], Sturgeon models are not patient-specific and can be used universally without retraining, mitigating the need to have access to privacy-sensitive training data at the site of deployment. As a result, limited computational resources are required during surgery. For example, the Sturgeon classifiers shown here can classify a Megalodon output file containing data from 32,610 reads in 17 s on an AMD Ryzen 7 6800H central processing unit. As the model inference step practically poses no constraint on the time it takes to classify a sample, it is possible to run multiple Sturgeon classifiers in parallel. Furthermore, we show that the models perform robustly across different sequencing flowcell types (MinION and PromethION, using R9 and R10 chemistry), laboratories (Utrecht, Amsterdam and Oslo) and methylation calling methods (Megalodon/Rerio, Guppy/Remora and Nanopolish). However, similar to other methylation-based classifiers, the performance of Sturgeon is limited by tumour purity in the analysed sample, and cannot account for intratumour heterogeneity.

We envision training of improved versions of Sturgeon as more data become available. The class definition used in the Capper et al. data[14] used for training Sturgeon has since been updated several times, with the addition of many new classes[32]. When these data or data from in-house cohorts are available, retraining and/or fine-tuning the Sturgeon model will be straightforward. Leveraging data from many different institutes across different countries for training machine learning algorithms is complicated owing to data-sharing restrictions as a result of privacy legislation that follows from patient consent. Sturgeon is ideally suited to address this as it can readily be employed in a federated learning setting[33]. Moreover, owing to the simulation approach used by Sturgeon, we envision that different types of training data, such as those obtained using other microarray platforms, nanopore or bisulfite sequencing can all be naturally accommodated.

Ultra-fast methylation sequencing holds great potential for several other fields of application. For routine (post-operative) diagnostics, turnaround times can be substantially reduced, reducing patient distress and anxiety and allowing tailored treatments to start as soon as possible. Furthermore, the low investment cost enables application in peripheral centres and centres with limited financial means. A longer-term future application may be to support administration of implantable therapies, which have the potential to bypass the blood–brain barrier. Current applications are associated with a high complication rate[34] and are so far limited to recurrent tumours[35], where the specific tumour type is known.

A potential limitation of the Sturgeon approach is the required amount of tissue. So far, we have instructed surgeons to obtain a sample measuring roughly $5 \times 5 \times 5$ mm, as this yields a high concentration of DNA suitable for library preparation. We have, however, successfully extracted sufficient DNA from smaller samples, including a biopsy. We note that—particularly when using the R10 workflow—as little as 200 ng of genomic DNA is sufficient. However, for specific applications—such as needle biopsies—more sophisticated (but slower) extraction protocols may be required to obtain sufficient DNA.

In conclusion, our results demonstrate that turnaround times of 1.5 h are feasible for the majority of samples. This is fully compatible with the surgical timeline, to guide the surgeon on how to proceed with the procedure. We envision clinical application of Sturgeon to be deployed in parallel to histological assessment by a trained pathologist, who then integrates the histological and molecular results into an improved intraoperative diagnosis. Using Sturgeon in this way could also reduce the requirement for a confidence score of at least 0.95, since the pathologist will always weigh the predicted tumour class in the context of the observed tumour histology, patient history and tumour location. Sturgeon can have an especially important role in guiding decision-making in challenging cases where the histological diagnosis is ambiguous.

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

## Methods

### Data simulation

Short nanopore sequencing runs yield sparse and random coverage of the genome. To enable model training, we generate simulated sparse nanopore runs based on microarray data. To this end, $N$ simulated reads are randomly sampled from the read length distribution ($D$) and assigned a start mapping position in the genome. $N$ and $D$ are defined based on an average nanopore whole genome sequencing run using a MinION flowcell (Fig. 1a). Forward or reverse direction is chosen at random (50% chance each). Reads are clipped at the start/end of the chromosome. Given this set of reads, the covered CpG sites are determined and their binarized methylation status is obtained from the microarray sample. Raw EPIC profiles were binarized with a cut-off of beta ≥0.6 using scripts kindly provided by the authors of nanoDx[22]. To include measurement noise due to sample heterogeneity, and methylation calling error rate, 10% of the covered CpG sites are randomly flipped. To reduce overtraining on specific sparsity levels, we simulate runs of different sparsity levels in a balanced manner (see 'Submodel training'). To ensure reproducibility and avoid simulation leakage between samples of the different cross-validation folds, simulations can be completely deterministic (with the exception of noise) given a random seed and the simulation time.

### Cross-validation

To assess model performance the Capper et al. dataset[14] is split in four equally sized class-stratified folds. Two folds are used for submodel training, one for validation to assess the best model state during training and to perform score calibration. The final fold is used for testing to assess the submodel performance. Folds are rotated so that a total of four submodels are obtained. Simulations are tightly controlled through the seeds of the pseudo-random number generator—that is, training, validation and test seeds are mutually exclusive—to avoid cross-validation leakage. We used seed values between 0 and 499 for the test fold, between 500 and 999 for the validation fold and between 1,000 and 1,001,000 for the training fold.

### Neural network architecture

Sturgeon (named thus to fit in the traditional fish-based nomenclature for nanopore software and because it sounds like 'surgeon') is a neural network containing three fully connected layers. The first two layers have 256 and 128 dimensions respectively, and are followed by a sigmoid activation (Extended Data Fig. 1). The first layer has an input size of 428,643, corresponding to the number of probes on the arrays used in training. The last linear layer has a dimensionality equal to the number of classes to be predicted (91 dimensions for the general classifier, and 30 dimensions for the brainstem-specific classifier). The outputs of the neural network are calibrated by a learned scalar value (see 'Score calibration'), and then transformed to probabilities via the softmax function. Dropout rate between layers was set to 0.5. As classification loss cross-entropy with uniform weights was chosen.

### Submodel training

We train the neural network as a supervised multi-class classification problem. Longer simulations contain more information, as more CpG sites are measured, and are therefore easier to classify; shorter simulations are more difficult. We therefore use a curriculum learning approach, where we first start by training the neural network with a mix of easy and difficult simulations, and later on move to train on only more challenging ones. We therefore first pretrain the neural network on the Capper et al. samples[14] (91 classes) using simulations that range between 0.6% and 14% sparsity (this range contains both easy and difficult to classify simulations). We then fine-tune this neural network for the final classifier by training using simulations that range between 0.6% and 6.3% sparsity (this range contains more difficult

to classify simulations). For the brainstem classifier, the last layer is substituted by an untrained layer with the correct dimensionality (30 classes).

We pretrain the neural network for a total of 3,000 epochs with a batch size of 256. For this purpose the AdamW[36] optimizer is used with a starting learning rate of $10^{-5}$ that increases linearly for the first 1,000 training batches until $10^{-3}$; afterwards, it is decreased using a cosine function until it reached $10^{-4}$ on the 1,000th epoch; we then keep training at a constant learning rate of $10^{-4}$ for 2,000 epochs. Other parameters of the optimizer are: $\beta_1 = 0.9$, $\beta_2 = 0.999$, $\varepsilon = 10^{-8}$ and $\lambda = 0.0005$. During training, we apply a dropout rate of 0.5 between all layers. We define one epoch as the number of reference samples in the most abundant class multiplied by the number of output classes. For every 2,000 training batches the current weights of the model are saved; and the model is evaluated on 50 validation batches (12,800 samples) by calculating their average loss and sensitivity. Validation batches are sampled in the same manner as the training batches, with the exception that simulation seeds are independent. We fine-tune the neural network using the exact same parameters as described for the pretraining, with the exception that we fine-tune for 3,000 epochs with a constant learning rate of $10^{-4}$.

During inference, we classify samples using the four trained submodels and use as final classification the scores from the model with the highest confidence. For the general, classifier we sum up the scores of two highly similar pilocytic astrocytoma classes (posterior fossa pilocytic astrocytoma and midline pilocytic astrocytoma into a single class low grade glioma pilocytic astrocytoma) and two highly similar medulloblastomas (*SHH* medulloblastoma child/adult and *SHH* medulloblastoma infant into *SHH* medulloblastoma).

### Adaptive sample balancing

Because of class imbalance in the training dataset, all classes are upsampled such that they are equally represented by simulating additional samples for classes smaller than the largest class. Similarly, we balance the sequencing sparsity levels such that the training data for each class consists of samples that have a uniform distribution of simulated sequencing times. At the end of each epoch, we recalculate the class balance by increasing the upsampling of classes and/or simulation times for which the model performs worse. Conversely, classes and/or simulation times for which the model performs well are upsampled relatively less. The number of samples for each class ($c$) and sparsity level ($t$) for epoch $i + 1$ is provided by:

$$\text{NumSamples}_{t,c}(i+1) = \text{TotalNumSamplesEpoch} \times \frac{\text{Error}_{t,c}(i) + 0.3}{\sum_t \sum_c \text{Error}_{t,c}(i) + 0.3}$$

We add a correction constant (0.3) to avoid completely removing classes or timepoints from the epoch. The total number of samples per epoch is kept constant (13,013 samples), based on the first epoch.

### Score calibration

To calibrate the classifier scores, enabling interpretation of these scores as confidence scores, we use temperature scaling[24]. To this end, we use each validation fold sample to create 500 simulated sparse samples (using all validation fold seeds) for all sparsity levels (between 0.6% and 14% sparsity). Given the whole reference dataset (2,801 samples), this results in 16,806,000 total simulations. Based on these simulated samples, we optimize the scalar temperature parameter, used for calibration, by minimizing the class-weighted cross-entropy between the temperature divided non-scaled logits and the correct class label. For this purpose we use the L-BFGS algorithm implemented in PyTorch with learning rate 0.01 and a maximum of 500 iterations. We evaluate the calibration of the model using ECE, a statistical measure that summarizes the difference between classifier accuracy (acc) and confidence (con). The ECE is defined as the bin-weighted average of the absolute

difference between accuracy and confidence on equally sized bins $B$ (here we use 10 bins).

$$\text{ECE} = \sum_{m=1}^{M} \frac{|B_m|}{n} |\text{acc}(B_m) - \text{con}(B_m)|$$

## Model evaluation
We assessed the final performance of the model on the left-out test fold samples. For this purpose, each sample was used to simulate 500 sparse nanopore runs across all 12 sparsity levels. In this way, each sample contributes 6,000 simulated samples to the test set. We report top1 and top3 F1 scores for each class individually across all sparsity levels, as well as average metrics across classes. F1 score, as an evaluation metric, is chosen since it considers both types of errors (false positives (FP) and false negatives (FN)), but it does only include true positives (TP), and not true negatives (since these would inflate the metric massively due to the multi-class setting). We also report the TPR for each class.

$$\text{F1 score} = \frac{2 \times \text{TP}}{2 \times \text{TP} + \text{FP} + \text{FN}}$$

$$\text{TPR} = \frac{\text{TP}}{\text{TP} + \text{FN}}$$

## Pseudotime
To reduce costs, some samples used for validation were sequenced on washed flowcells, or multiple samples were multiplexed on a single MinION or PromethION flowcell. Re-used flowcell and multiplexed sequencing times are not directly comparable to sequencing runs of a single sample on a new flowcell. Similarly, samples sequenced on PromethION flowcells are not directly comparable to MinION flowcells due to their larger throughput. In order to make these runs comparable to a real intraoperative scenario (one sample sequenced on a new MinION flowcell), we use the number of CpG calls as a proxy for sequencing time. For this purpose, we first estimate the expected sequencing throughput in terms of the median number of CpG sites covered per 5 min time interval from a collection of 6 representative MinION runs that used fresh flowcells (Supplementary Fig. 12). The number of CpG sites (non-cumulative) covered in the first 12 intervals of 5 min are: 51,924; 104,073; 124,078; 149,111; 173,504; 194,399; 207,456; 217,193; 232,101; 241,278; 247,600; and 258,197. Thus, for a multiplexed (MinION or PromethION) or washed flowcell sequencing run, we assume a fresh flowcell equivalent: 5 min of sequencing is achieved when 51,924 CpG sites are covered. By using throughput, instead of number of reads, this allows us to properly simulate the ramp up in sequencing throughput that happens during the first minutes of sequencing.

## Robustness to random sampling
To further analyse the robustness of Sturgeon, we create additional realistic nanopore sequencing data by randomizing the order of the sequenced reads. We randomize the sequenced read order of each sample 100 times and evaluate at which pseudotime the desired threshold would have been reached and whether the classification was correct or not.

## Robustness to sample purity
To analyse the robustness of Sturgeon to impure samples, we simulated nanopore runs with a mix of reads from tumour and control reference samples at different impurity levels. To achieve this, we simulated nanopore runs that contain between 5% and 95% control tissue reads in 5% increments. For each sample 100 independent nanopore runs were

simulated, containing reads counts equivalent to between 10 to 40 min of sequencing. This produced a total of ~20 million simulated sequencing runs. We note that the tumour reference samples are not 100% pure, and contain some levels of non-tumour tissue. We therefore report the in silico purity by multiplying the original tumour fraction with the fraction of reads that are simulated from the tumour sample. We evaluate Sturgeon performance by cross-validation (see 'Cross-validation'), that is, each Sturgeon submodel was evaluated on simulations from samples and simulation seeds in the test fold.

## Paediatric methylation profile validation
At the PMC centre for paediatric oncology, EPIC arrays are routinely performed on paediatric CNS cancer samples. We gathered 94 such profiles that were generated in the routine diagnostic process. Raw EPIC profiles were binarized with a cut-off of beta ≥0.6 using scripts kindly provided by the authors of nanoDx[22]. EPIC probes not present on the 450K array used in the reference cohort were filtered out. We then simulate 500 nanopore runs at 12 sparsity levels as described above. The EPIC profiles were all submitted to the Heidelberg v11b classifier (with the exception of PMC_20 which was classified with classifier v12.5), results (Sturgeon classification result and confidence score) are listed in Supplementary Table 3. Samples were also labelled with a 'final diagnosis', the result of a combination of histological assessment, imaging, CNV profiling and molecular characterization which we consider the ground truth.

## Classification of publicly available nanopore sequencing data
We downloaded nanopore sequencing data from GSE209865 (ref. 22). Of note, this dataset consists of processed sequencing data, which uses a different processing method consisting of Guppy (v3.1.5) base calling, Nanopolish methylation calling, and mapping to hg19. This dataset consists of binary methylation calls for 450K methylation sites and can thus directly be used for Sturgeon classification.

## DNA extraction
DNA is extracted from (fresh/unfixed) tumour samples using an adapted QiaAmp mini (Qiagen) protocol. Ideally, a tumour sample of roughly $5 \times 5 \times 5$ mm is used as input material. ATL buffer (180 µl) is added to the sample and the sample is briefly ground using a pestle, then 200 µl buffer AL and 20 µl proteinase K are added and the sample is moved to a 70 °C hea block. Once heated the sample is ground with a pestle every minute to improve proteinase K accessibility. When the sample contains no more solid tissue, or after 5 min of incubation and grinding, the sample is added to a Qiashredder column (Qiagen ID: 79656), not including any solid matter if still present. The column is centrifuged at 20,000$g$ for 1 min. 200 µl of 96% ethanol is added to the eluate and the eluate is moved to a qiaAmp column and centrifuged for 1 min at 6,000$g$. The column is washed with 500 µl AW1, centrifuged at 6,000$g$ for 1 min, then with 500 µl AW2 at 12,000$g$ for 1 min. The remaining ethanol is removed in a fresh elution column, centrifuged at 12,000$g$ for 10 s. Sample is eluted with 25 µl of MilliQ water. Samples are quantified using a Nanodrop spectrophotometer (Thermo Fisher). For the first few intraoperative cases, we encountered samples where the tumour cell purity was low, resulting in low confidence scores, or high confidence scores for control tissues. As there is very limited time between sampling and processing, the tumour cell content cannot be rigorously assessed prior to DNA isolation. Instead, after the fourth case, where possible, we implemented an approach where DNA from up to three distinct sections of the sample was isolated. Simultaneously, the pathologist assesses tumour content from these same sections and calls the DNA isolation laboratory to report with which sample to continue. This procedure only slightly delays the process (three samples are processed instead of one during the DNA isolation), and the pathologist is usually able to relay this information before DNA isolation is completed.

## Flowcell chemistry versions

During the collection of samples, we migrated from R9 MinION flowcells (which are being discontinued) to R10.4.1 MinION flowcells. Since Sturgeon uses a list of CpG sites and their binary methylation state as input, we do not expect nor observe an effect on classifier performance. This was also confirmed by re-sequencing and processing five samples on an R10.4.1 MinION flowcell that were previously sequenced using an R9 PromethION flowcell. Importantly, we observe increased throughput on R10.4.1 flowcells, in the range of 1,000–1,200 reads per minute and slightly higher concordance between methylation array methylation calls and nanopore sequencing methylation calls (Supplementary Table 1 and Supplementary Fig. 12).

## Library prep

Samples are library prepped depending on whether an R9 or R10 flowcell is to be used. For R9 flowcells, the Oxford Nanopore RBK004 kit was used, using 600 ng input material and following manufacturer's instructions for other steps. For R10 flowcells, we used the Oxford Nanopore technologies RBK114-24 kit, but with an adjusted protocol: for optimal results (size distribution centred around 5 kb) 3,500 ng of input material in 50 µl is first sheared using a G-Tube (Covaris SKU: 520079), centrifuging at 7,200 RPM (6,000g) in a fixed rotor tabletop centrifuge. Subsequently 7.5 µl of the sheared input material is used for tagmentation with 2.5 µl indexing mix. Alternatively, we obtain similar results (but with wider size distributions) using 200 ng input material for the tagmentation without an added fragmentation step. For both protocols, we omit the AMPure purification, and after tagmentation directly proceed with adapter ligation.

## Flowcell loading

ONT MinION sequencing initializes with a pore scan, which takes around 5 min and produces no reads. Therefore, with R9 flowcells, we start the sequencing as soon as the sample arrives in the laboratory, so that sequencing commences as soon as the library is loaded onto the flowcell. Flowcells are primed using 800 µl Flush Buffer (from the ONT flowcell priming kit) at the start of the DNA isolation, after five minutes the flowcell is flushed with 200 µl Flush Buffer and sequencing is started, at which point the software will first perform a pore scan. The DNA library is loaded as soon as it is ready. By then the pore scan has typically finished and sequencing commences. We noted that this procedure has an adverse effect for R10 flowcells, and for these we only start sequencing after loading the library on the flowcell.

The sequencing itself is slowed down by the startup phase (Supplementary Fig. 12), where many unligated sequencing adapters are sequenced, and then ramps up towards higher pore activity and more informative reads per minute (usually stabilizing in the range of 800–1,200 reads per minute). We typically obtain 10,000–20,000 reads within 1 h after the sample arrives in the isolation laboratory. In some, but not all, cases this is enough for a reliable diagnosis. After an 90 min we typically reach 40,000–60,000 reads. After sequencing flowcells were routinely washed using the EXP-WSH004 flowell wash kit (Oxford Nanopore Technologies) according to the manufacturer's instructions and stored for later use.

## Methylation calling

To call methylation from R9 chemistry nanopore data we used Megalodon V2.5.0, which runs with Guppy V5 to perform base calling and mapping to the CHM13V2 reference genome. To call per-read-per-site methylation we use the Rerio CpG methylation model as described[23] We convert the methylation log likelihood ratio to a probability and use a cut-off of <0.3 for unmethylated and >0.7 for methylated. For R10 chemistry, we use Guppy V6.4 and the high-accuracy CpG methylation calling model to call methylation and again use the <0.3 and >0.7 cut-offs to call methylation.

Methylation calls are assigned to one of the 450K CpG sites present on the Infinium methylation array using windows centred on the CpG site targeted by each probe. We benchmarked different window sizes and observed that for R9 chemistry (and associated methylation calling procedure), 100-bp windows provide optimal results. With R10 chemistry, smaller windows do not reduce the methylation calling accuracy much, but for simplicity we also opted to use a 100-bp window (Supplementary Table 1). If multiple CpG sites are present within the window, majority voting is used to convert the calls to a single call per site. When multiple reads cover the same Infinium probe site, majority voting is also used to create one methylation call. When voting results in a tie, we discard that particular site. The methylation calling error rate was evaluated by comparing the methylation calls between nanopore sequencing and the microarray data for the same samples where both methods were available. This indicated a concordance of 85–90% between binarized array data (beta cut-off at ≥0.6) and nanopore methylation calls (Supplementary Table 1). The error rate is evenly distributed between false positives (calling unmethylated sites as methylated) and false negatives (calling methylated sites unmethylated).

## CNV calling

The CNV calling approach was adapted from Kuschel et al.[22] Even with several hours of sequencing, coverage is still sparse (50,000 reads equates to 1 read per 64 kb of genomic sequence). Therefore, we restricted the analysis to genomic bins spanning 2 Megabases using the QDNAseq package[37]. We normalized the coverage using a publicly available deeply sequenced Genome In a Bottle reference sample (NA12878 release 7; https://github.com/nanopore-wgs-consortium/NA12878/blob/master/Genome.md), to correct for nanopore sequencing specific mapping biases against the CHM13V2 reference genome. To obtain a log ratio, we first calculate the relative coverage (sample coverage/reference coverage) per bin, after which the $\log_2$ of the relative coverage over the mean relative coverage is taken. We apply the DNAcopy[38] R package to segment the genome into over or underrepresented regions, indicating potential CNVs.

## Live analysis

We developed a custom R script (available on the Sturgeon github) to run parallel to the sequencing software. We run MinKNOW (v22.12.7) with disabled base calling. MinKNOW outputs fast5 files, each containing 4,000 reads. The R script checks the MinKNOW output folder for a new fast5 file, and triggers if it contains 4,000 reads. Complete fast5 files are copied to a separate working directory where they are processed using either Megalodon or Guppy depending on the chemistry version. For R9 chemistry, qCat V1.1.0 (https://github.com/nanoporetech/qcat) is used to identify barcodes and for R10 chemistry the Guppy built-in barcoding detection algorithm is used. Depending on user settings, either the most frequent or a user-specified barcode is selected. Per-read-per-site methylation calls (Guppy v5.0.1 and Megalodon v2.5.0) or.bam files (Guppy v6.5.2) originating from reads with the selected barcode are saved and Sturgeon is used to map the methylation calls to the 450K CpG sites and classify the sample. Sequencing and analysis are performed on an ASUS TUF A15 FA507RR-HN003W laptop with 64 Gb RAM. For R10 sequencing experiments we use Guppy v6.4.6 with the dna_r10.4.1_e8.2_400bps_modbases_5mc_cg_hac.cfg configuration file, with disabled Q-score filtering and minimum barcode quality set at 6. As we are using single samples, demultiplexing is not strictly necessary and we process reads with and without classified barcodes.

## Adaptive sampling

Adaptive sampling can be performed in exclusion or enrichment mode, and can be configured either through specifying a reference genome and a.bed file or a.fasta sequence file. Additionally, buffer regions can be specified, which, in our case, determines how much flanking sequence is accepted around the CpG sites of interest. We performed

simulations of different buffer regions assuming the read length distribution of earlier sequence experiments. This indicated that 5-kb buffer regions are optimal (data not shown). Together, these windows span 1.3 billion basepairs (approximately 40% of the genome). We then designed both.bed file and.fasta files for 5-kb flanking regions, merging any regions that are within 25 kb using Bedtools merge (Bedtools V2.30)[39]. When choosing between adaptive enrichment and adaptive exclusion, we opted for adaptive enrichment, as adaptive exclusion will read up to 4,000 bp before deciding to exclude a read, whereas adaptive enrichment decides after a maximum of 400 bp. Finally, initial experiments showed higher efficiency when using a reference genome and a.bed file with regions of interest compared to using a.fasta file with target sequences. We assessed the added value of adaptive sampling by running 5 samples on MinION R10.4.1 flowcells where adaptive sampling was enabled for half of the available channels. This allowed us to make a fair comparison between adaptive and non-adaptive sequencing on the same flowcell and library. All adaptive sampling experiments were performed on an Oxford Nanopore GridION device running MinKNOW 22.12.5 using high-accuracy base calling.

### Consent for publication

The research was approved by the Biobank and Data access committee (BDAC) of the Princess Máxima Center for paediatric oncology. All included patients provided written informed consent for participation in the biobank (International Clinical Trials Registry Platform: NL7744; https://onderzoekmetmensen.nl/en/trial/21619). Inclusion of intraoperative samples was ethically approved via the same decision; the results were not shared with caregivers and therefore not used to alter patient treatment nor diagnosis in all but two cases. In these two patients with paediatric CNS tumours (included in the very last phase of this study) the result of intraoperative nanopore sequencing analysis was shared with the neurosurgeons during operation; in these cases the result were ependymoma (subtype *RELA* fusion, and type A), corroborating and fine-tuning the provisional intraoperative frozen-section diagnosis 'ependymoma' that was already communicated with the neurosurgeon at an earlier phase of the operation. For patients at the Amsterdam University Medical center with glioma, a broad consent was signed for use of material to improve clinical methods. For patients at the AUMC we publicly share methylation calls but not raw DNA sequence data as there is no explicit consent for sharing genetic information.

### Patient identity

To protect patient privacy, patient IDs were generated and assigned exclusively for this publication for patients included from the Princess Máxima Center and the Amsterdam University Medical Center. The translation between patient IDs and patient identity is not known outside this research group. For publicly available data, the identifiers were maintained from the metadata provided by the repository.

### Reporting summary

Further information on research design is available in the Nature Portfolio Reporting Summary linked to this article.

### Data availability

Paediatric patient data are available through the European Genome-Phenome Archive EGAS00001007475, data from patients at AUMC are accessible through NCBI Gene Expression Omnibus (GEO) with accession code GSE237874. Data used for training are available at GEO under accession number GSE109381.

### Code availability

Code used to train and validate the model can be found at: https://github.com/marcpaga/sturgeon_dev[40]. Source code of the Sturgeon prediction tool as a Python package can be found at https://github.com/marcpaga/sturgeon, together with links to download the two (general and brainstem) trained models used in this work[41]. The following Python v3.7 packages were used during the development of Sturgeon: torch v1.10.2, numba v0.56.2, onnxruntime v1.12.1, scipy v1.7.3, scikit-learn v1.0.2, pandas v1.3.5, numpy v1.21.5, matplotlib v3.5.1, pysam v0.19.0, modbampy v0.6.3. The following R v4.1.2 packages were used: QDNAseq v1.30.0, DNAcopy v1.68.0, rhdf5 v2.38.1. We also used Samtools v1.13 and bedtools v2.30.

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

**Acknowledgements** We acknowledge the Utrecht Sequencing Facility (USEQ) for providing sequencing service and data. USEQ is subsidized by the University Medical Center Utrecht and The Netherlands X-omics Initiative (NWO project 184.034.019). We thank the Princess Máxima Center Biobanking facility and Big Data Core for storing and providing samples and data. We also thank the UMC Utrecht Bioinformatics Expertise Core (www.ubec.nl) for the help in applying research data management according to the FAIR Principles. We thank W. de Leng and E. van der Biezen for retrieving data. We thank Cyclomics for allowing us to use their GridION device. this project was partially funded by the Oncode Institute technology development fund and a stichting Kinderen Kanker Vrij (KIKA) pilot project grant.

**Author contributions** C.V., M.P.-G. and J.d.R. designed and created the Sturgeon algorithm. C.V., M.P.-G., B.B.J.T., L.K., P.W. and M.E.G.K. designed and performed practical implementation of intraoperative sequencing. B.B.J.T., L.K., E.J.K. and L.D. collected and managed patient data. M.E.G.K. and P.W. performed intraoperative frozen-section diagnosis and together with J.v.d.L. provided the final integrated diagnosis. K.v.B., E.W.H., N.V. and P.d.W.H. performed resection surgeries and provided samples. C.V., M.P.-G., B.B.J.T. and J.d.R. wrote the manuscript with input from all other authors.

**Competing interests** J.d.R., M.P.-G. and CV are inventors on a patent covering the development of Sturgeon. J.d.R. is co-founder and director of Cyclomics, a genomics company. L.K., M.E.G.K., P.W., N.V., P.d.W.H., E.J.K., L.D., J.v.d.L., K.v.B., E.W.H. and B.B.J.T. declare no competing interests.

**Additional information**
**Correspondence and requests for materials** should be addressed to B. B. J. Tops or J. de Ridder.

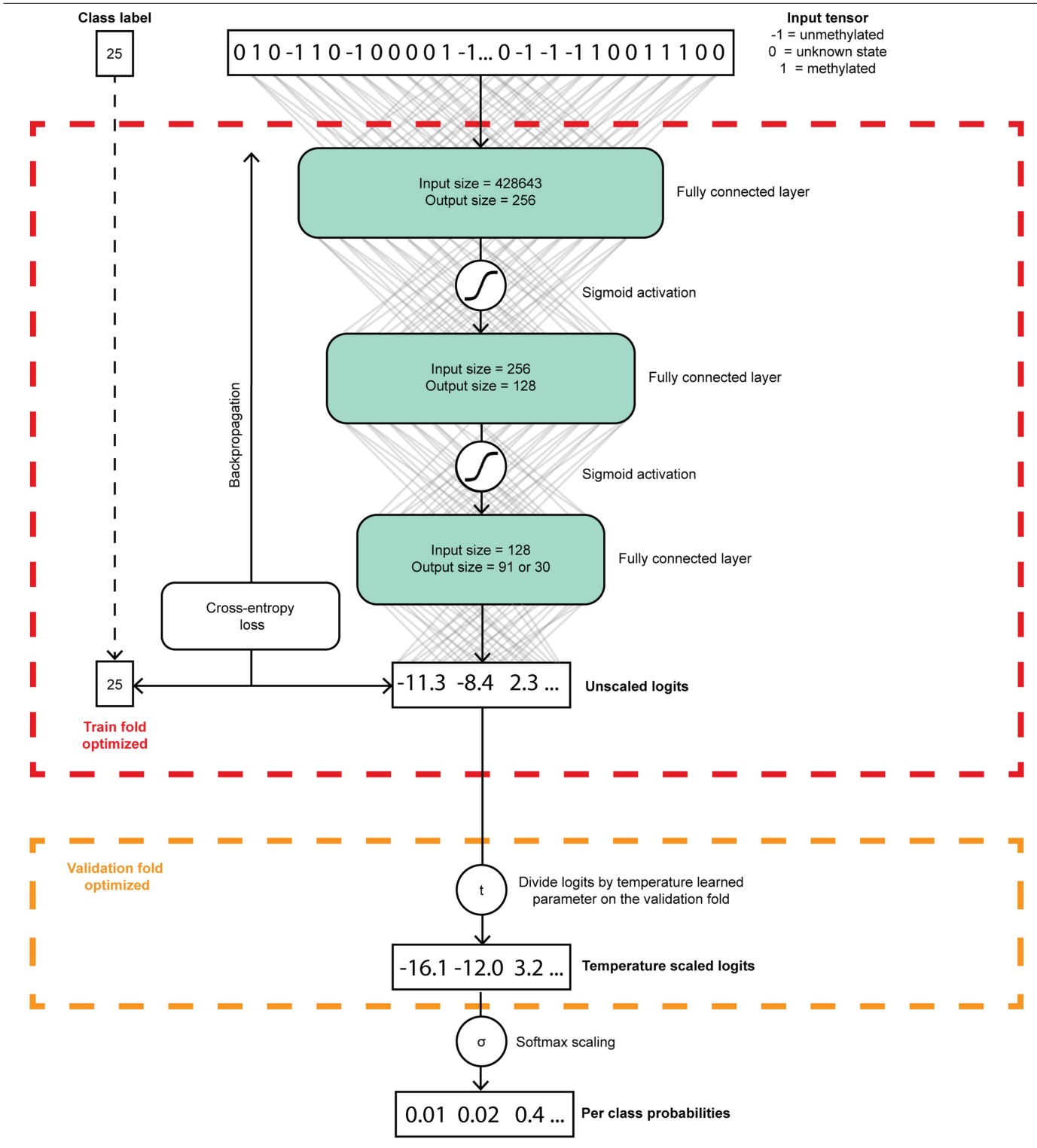

**Extended Data Fig. 1 | Neural network architecture and optimization scheme.** This schematic representation shows the architecture of Sturgeon neural networks. Each network consists of one input layer and three fully connected layers. The input layer represents the 428,643 CpG sites as individual nodes. The two hidden layers consist of 256 and 128 nodes and in the final layer each tumor class is represented as an output node. Validation folds are used for temperature scaling and a final softmax scaling is applied to scale the sum of all output nodes to 1.

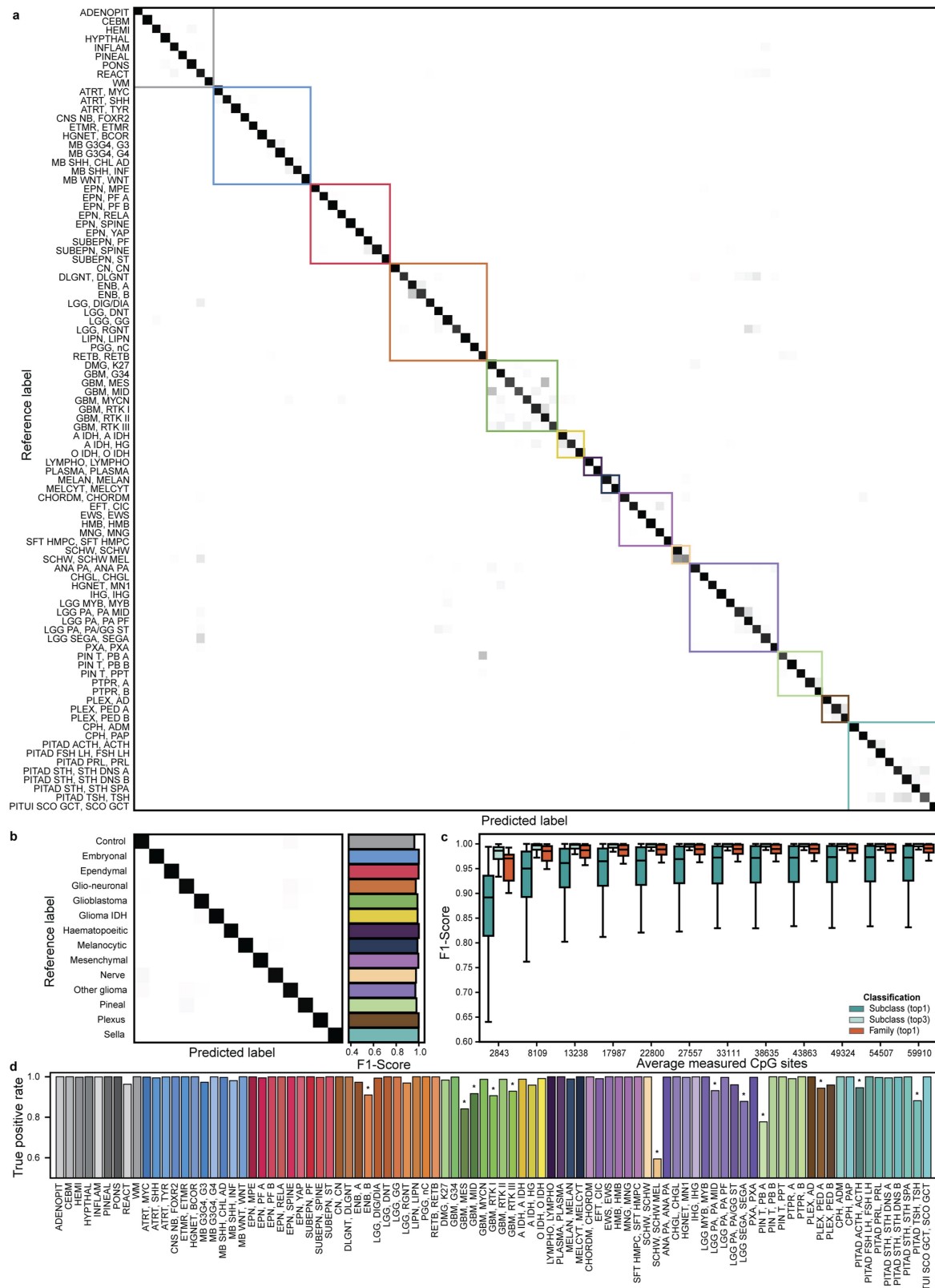

**Extended Data Fig. 2 | Sturgeon performance at 40 min of simulated sequencing. a**, Confusion matrix showing the highest scoring class for each reference label at 40 min of simulated sequencing (~97% missing values from microarray data) **b**, Confusion matrix and F1 scores at 40 min of simulated sequencing when scores are aggregated on the family level. **c**, F1 scores at different sequencing depths (represented by the average number of covered 450 K array methylation sites) when classifying by subclass, by the correct

subclass being in the top 3 of highest scoring classes and at the family level. Box plot minimum and maximum bounds represent the 25th and 75th percentiles, respectively, and the center bound represents the median. Whiskers extend to 1.5 times the interquartile range **d**, True positive rate for each subclass at 40 min of sequencing at the 0.95 confidence threshold. Asterisks indicate subclasses that do not reach the 0.95 true positive rate.

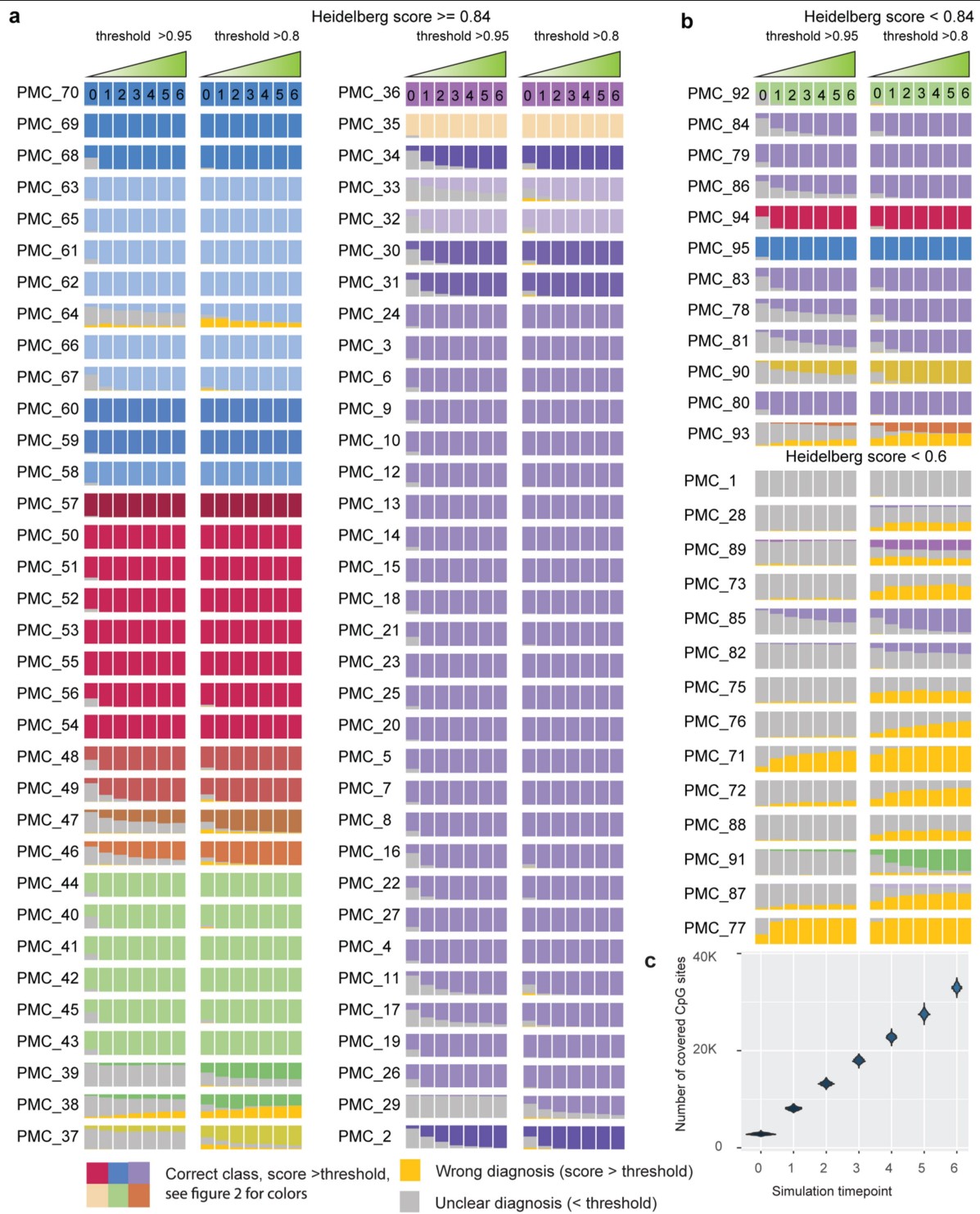

**Extended Data Fig. 3 | Classification performance over time on nanopore runs simulated from pediatric CNS tumor methylation arrays.** For each of 96 methylation profiles, a series of nanopore sequencing experiments were simulated. At each timepoint 500 experiments were simulated corresponding to approximately 5 min of sequencing per timepoint. Each bar indicates a consecutive timepoint and simulated sequencing data is accumulated over time. A stacked bar graph is plotted based on the number of correct, unclear or wrong classifications. Correct classifications are those with a confidence score >0.95 (left) and >0.8 (right) and with a class corresponding to the true diagnosis (bars are colored according to the class label). Unclear classifications are those with confidence-scores <0.95 or <0.8 colored in gray). Wrong classifications are misdiagnoses where a confidence-score >0.95 or >0.8 is obtained for the incorrect class (colored in yellow). **a** Clear diagnosis group (Heidelberg classifier score >0.84). **b** Difficult diagnosis group (Heidelberg classifier score <0.84). **c** Distribution of the number of CpG sites covered at each simulated timepoint.

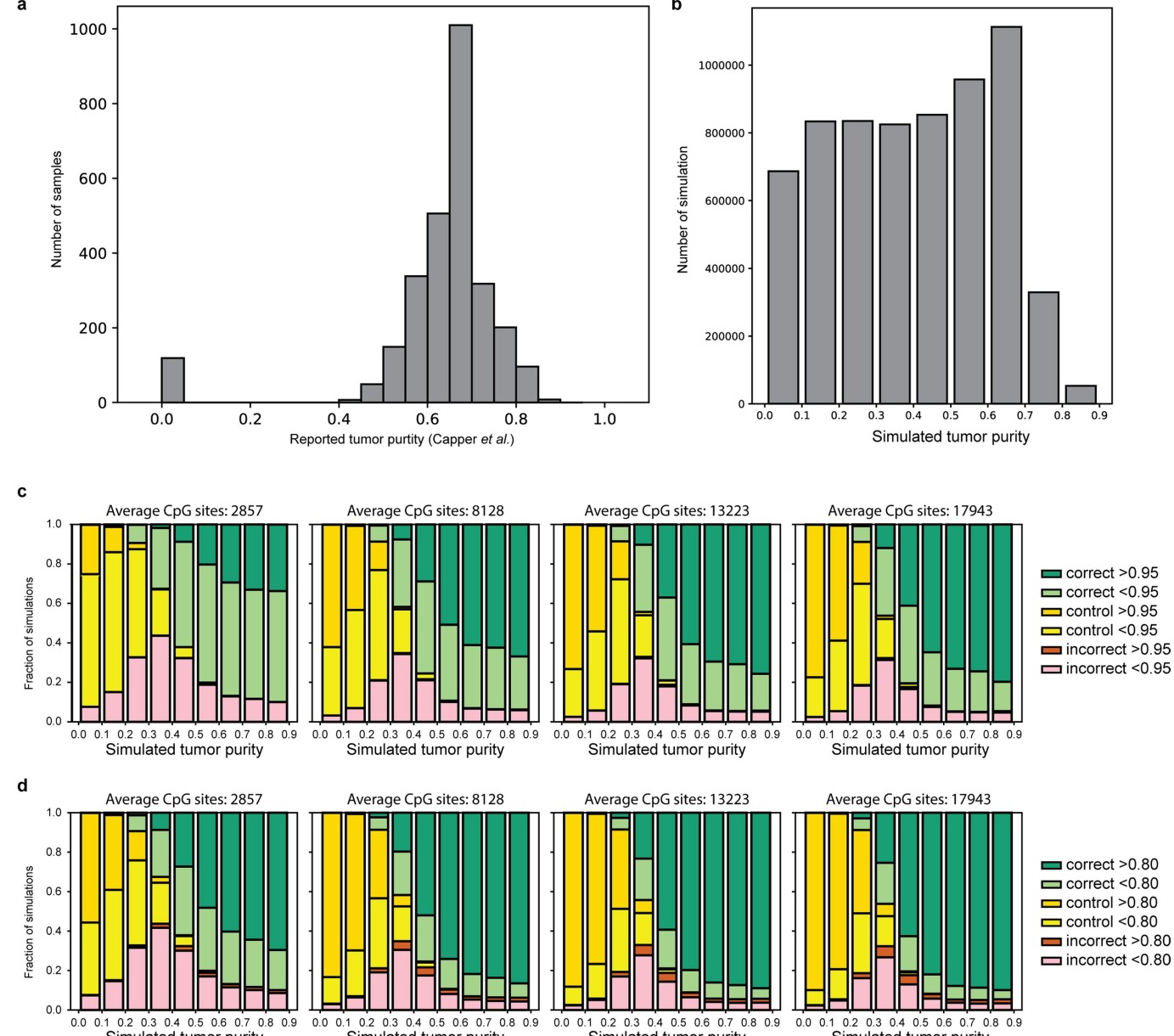

**Extended Data Fig. 4 | Sample purity simulation results. a**, Histogram showing the reported sample purity in the Capper *et al.* training dataset. **b**, Due to the inherent sample purity, the number of samples where high purity can be simulated is limited. This histogram shows the number of used simulations at each purity level. **c** and **d** barplots showing the simulation results at a 0.95 (**c**) or 0.8 (**d**) cutoff at different sequencing depths (represented by the average number of 450 K CpG sites covered.). Bars are colored by correct and confident (score above cutoff) outcomes, correct but low confidence outcomes (highest scoring class is correct, but the score is below the confidence threshold), high and low confidence control outcomes (the highest scoring class is one of the control classes), and wrong outcomes where an incorrect class scores highest below or above the confidence threshold.

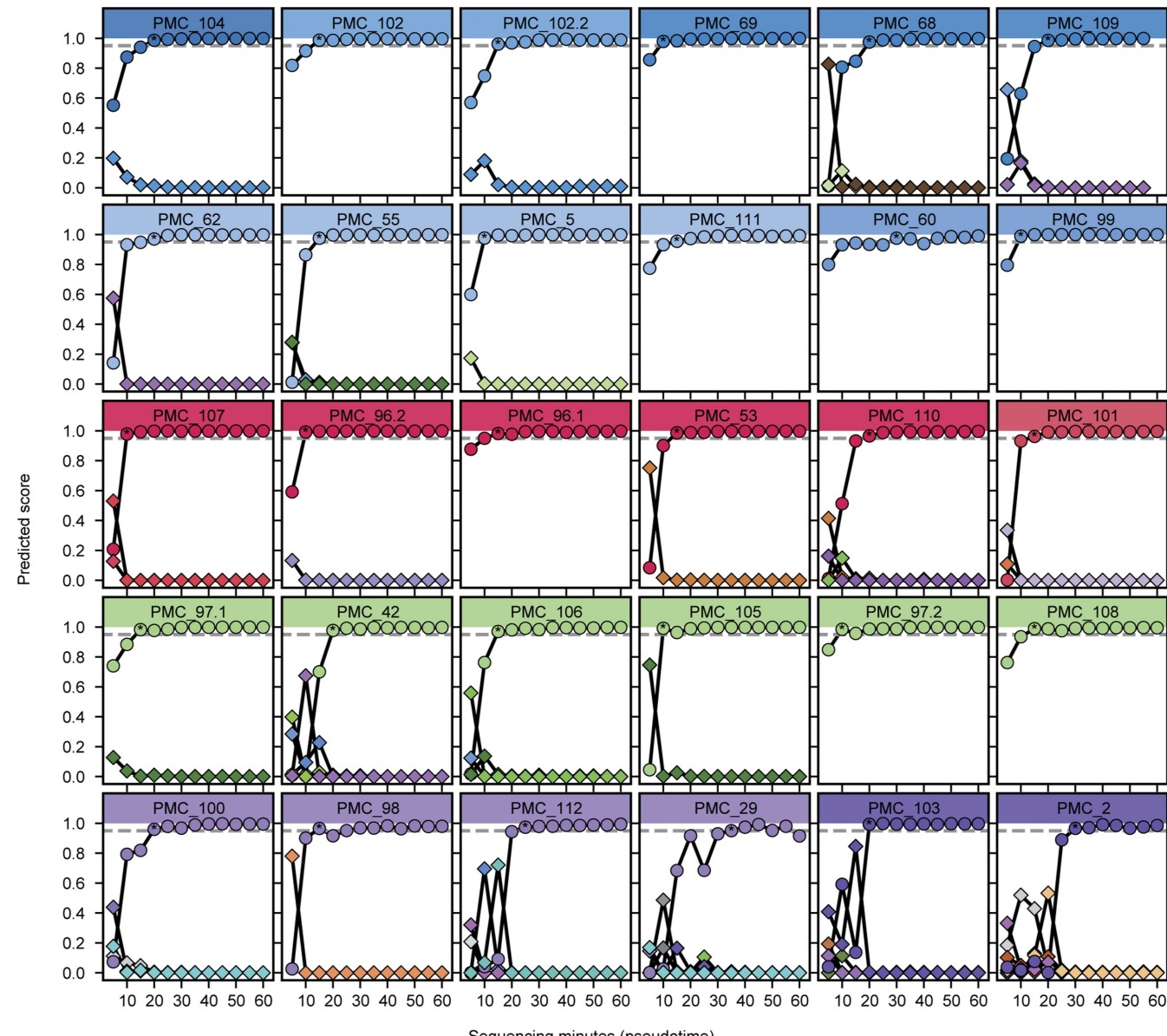

**Extended Data Fig. 5 | Retrospective nanopore sequencing results.**
Sturgeon confidence scores for 27 pediatric CNS tumor samples (duplicates indicated by appended "_1" to the sample name at increasing sequencing time (5 min pseudo time intervals). Top bar indicates the sample name. Circles indicate the predicted score of the correct class; diamonds indicate the predicted score of incorrect classes (classes with overtime averaged scores lower than 0.1 are omitted). Asterisks indicate the first time point where the score of the correct class was higher than 0.95. Horizontal line indicates the 0.95 threshold.

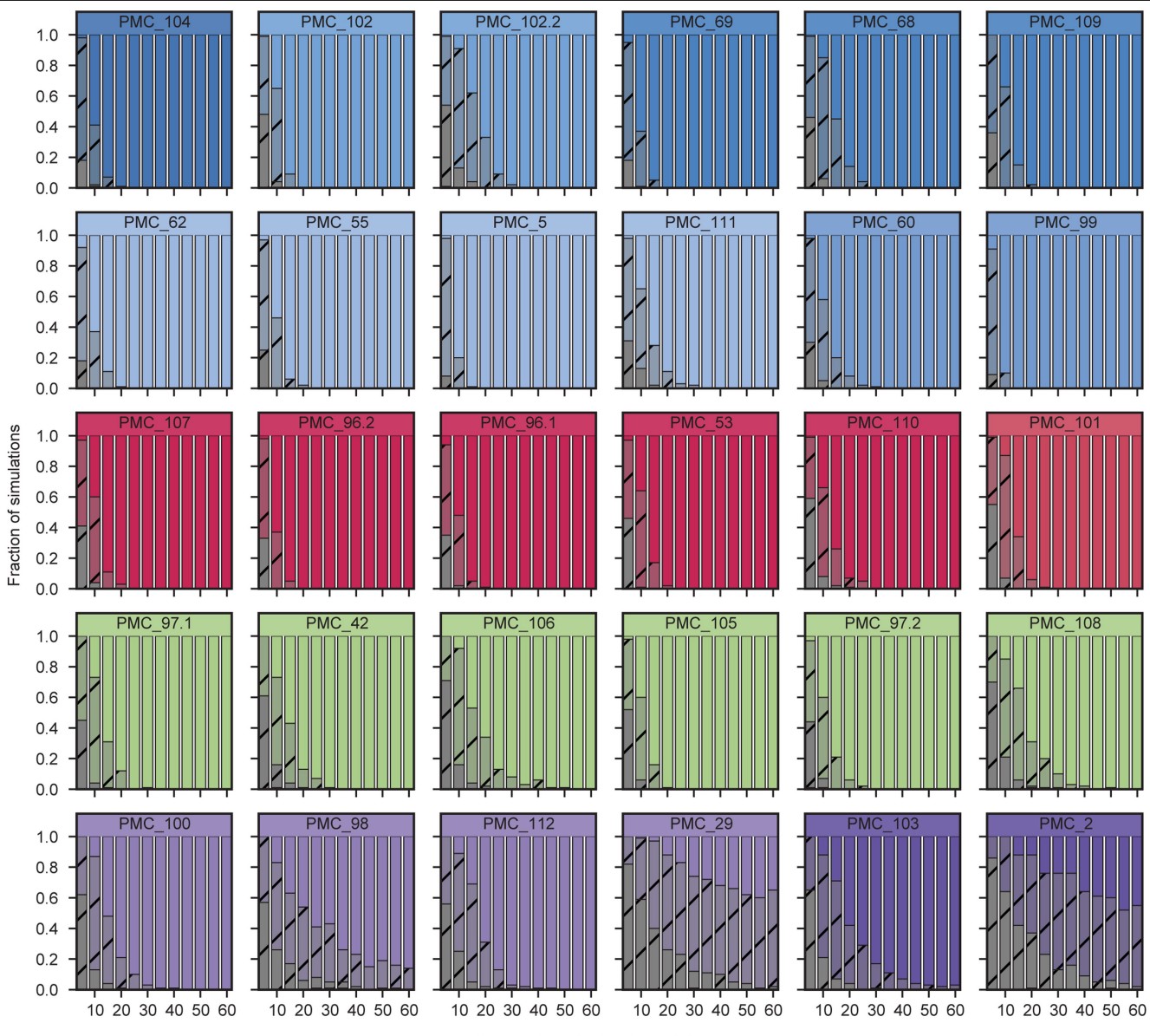

**Extended Data Fig. 6 | Robustness analysis results.** For each sample sequence reads were randomly sampled to reflect a nanopore run at a specific duration. 100 simulations were generated for each timepoint. Colored bars indicate correct outcomes above the confidence threshold (0.95), dashed colored bars indicate correct outcomes below the threshold, gray dashed bars indicate unclear outcomes and black bars indicate wrong outcomes above the treshold.

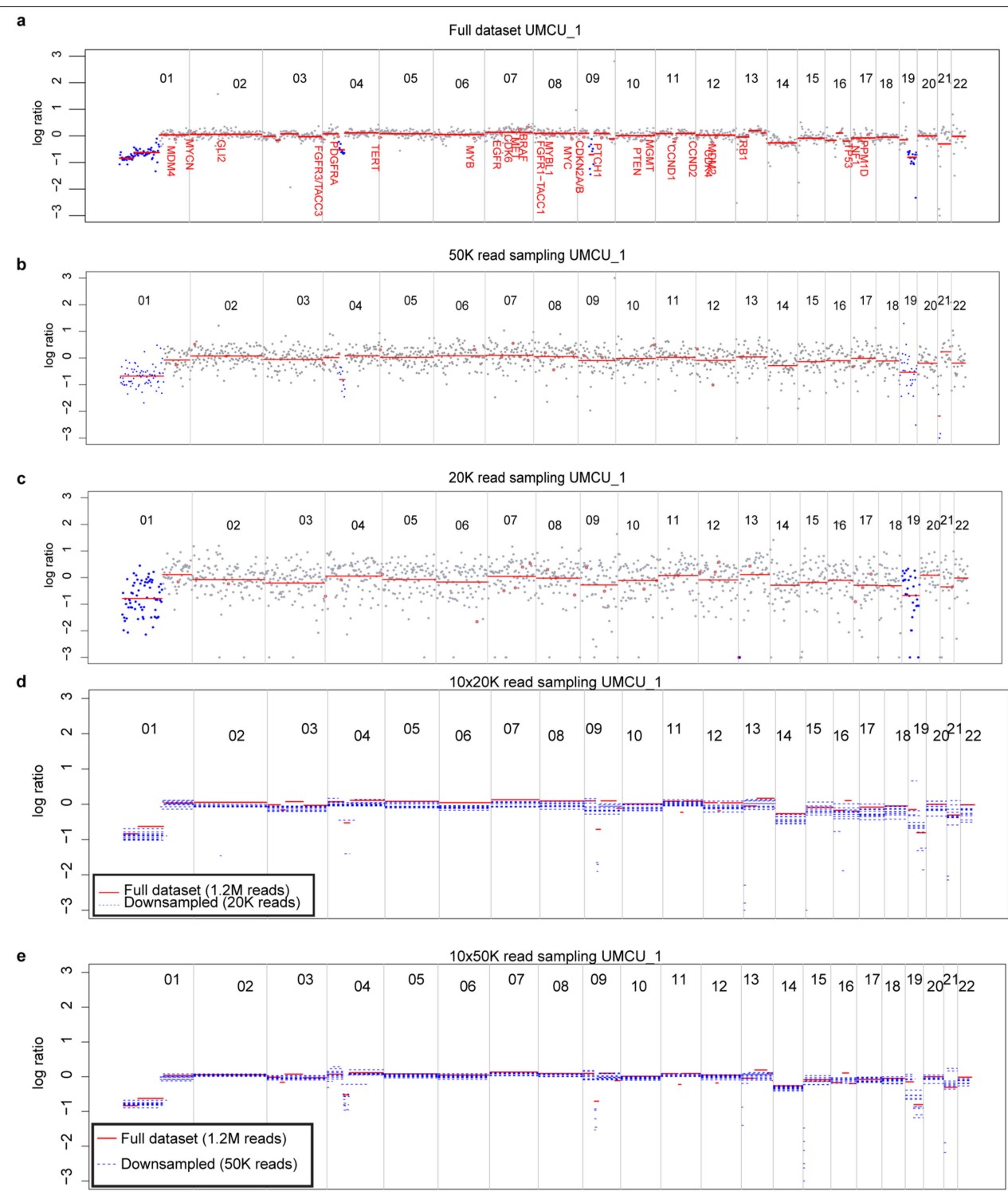

**Extended Data Fig. 7 | Copy Number Variations.** As a proof of principle we sequenced a retrospective Oligodendroglioma sample with a known 1p/19q codeletion. This sample was difficult to specify as an Astrocytoma or Oligodendroglioma based on methylation profile and histology (Supplementary Fig. 13). In such cases the 1p/19q codeletion offers strong supporting evidence for the Oligodendroglioma diagnosis. **a**, We sequenced this sample to 1.2 million reads, dots represent the normalized coverage (Methods) for 2 Mb bins, red lines indicate the DNAcopy segmentation result which clearly shows the 1p/19q

codeletion. Bins that fall within segments with a log2 value < −0.5 are colored blue and bins that fall in segments > 0.5 are colored green. **b** and **c**, We then subsampled to 50,000 and 20.000 random reads and repeated the analysis; in both sequence depths the 1p deletion is clearly visible, and the 19q deletion is visible but less clearly defined. **d** and **e**, the segmentation results from 10 random downsamplings at a sequence depth of 20.000 and 50.000 sequence reads respectively. Red lines indicate the segmentation of the full dataset and blue dashed lines show the result of individual subsamplings.

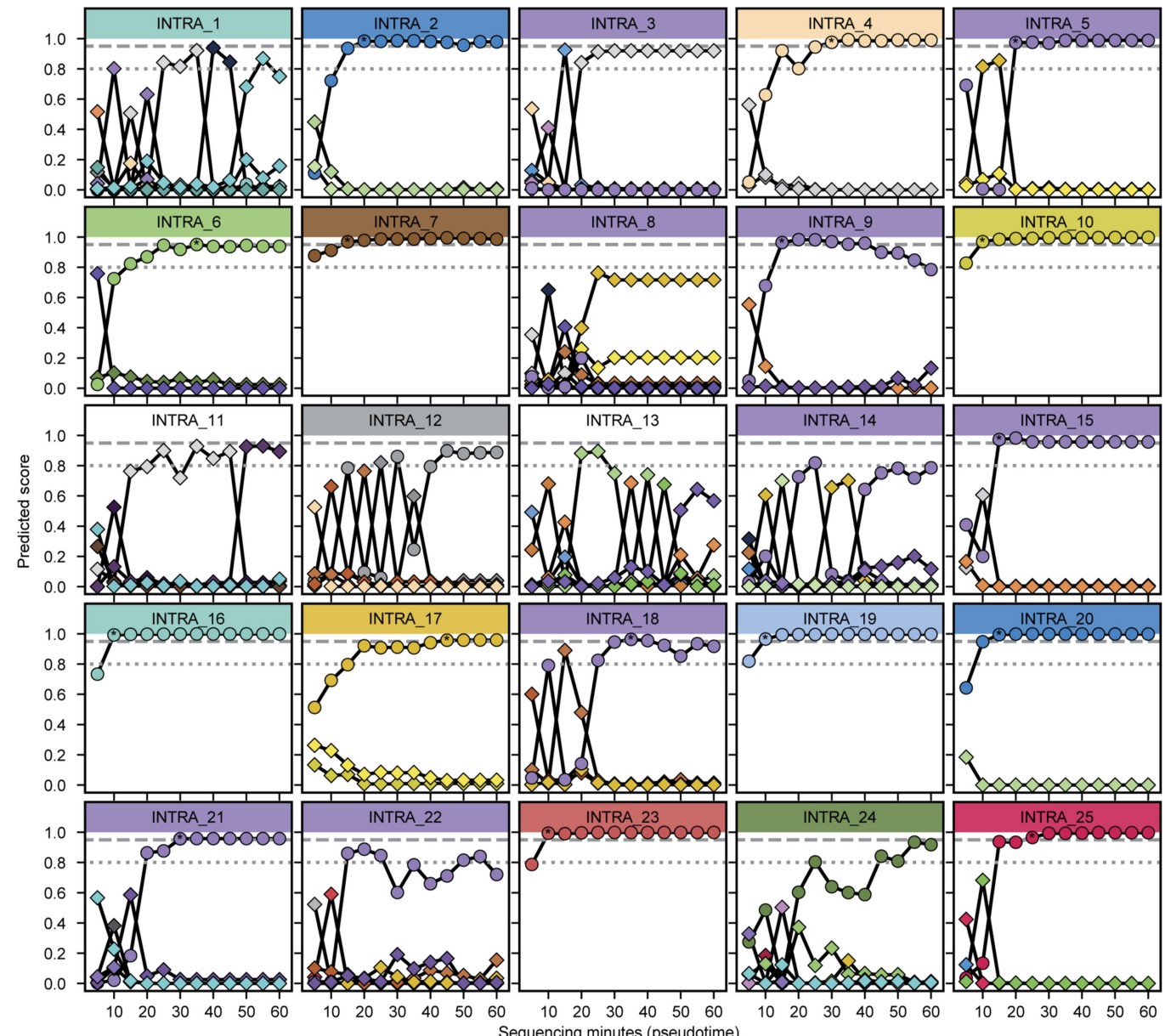

**Extended Data Fig. 8 | Intraoperative sequencing results.** Sturgeon confidence scores over time for the 25 intraoperative sequencing experiments. Class corresponding to the integrated histomolecular diagnosis are shown in circles (with the exception of INTRA_24, where the highest scoring Heidelberg V11b4 class is indicated as a circle) and other classes are shown as diamonds. Headers are colored following the same style as described in the circles, with the exception of INTRA_11 (Germinoma, class not in the classifier) and INTRA_13 (exotic case) which are colored white.

**Extended Data Table 1 | Overview of the intraoperative sequencing cases**

| | Sturgeon | CV | ST | Heidelberg | Diagnosis | Notes |
|---|---|---|---|---|---|---|
| 1. | Inconclusive | R9 | - | Not performed | Adamantinomatous Craniopharyngioma | *Sample used for intraoperative sequencing not representative. Diagnosis based on histology* |
| 2. | Medulloblastoma, non-*WNT* / non-*SHH* subtype: group 3 | R9 | 20 | Medulloblastoma, non-*WNT* / non-*SHH* subtype: group 3 (0.99) | Medulloblastoma, non-*WNT*/non-*SHH* | |
| 3. | Inconclusive | R9 | - | Pilocytic Astrocytoma, subtype: low grade glioma (0.59) | Pilocytic Astrocytoma | *Sample used for intraoperative sequencing not representative (blood clot). A later sample was used for array and integrated diagnosis* |
| 4. | Schwannoma | R9 | 30 | Schwannoma (0.99) | Schwannoma | *Process captured on film. (https://zenodo.org/record/8261128)* |
| 5. | Pilocytic Astrocytoma | R10 | 20 | Midline Pilocytic Astrocytoma (0.99) | Pilocytic Astrocytoma | *3 samples collected, two with low tumor percentage. Highest tumor percentage sample used in intraoperative sequencing* |
| 6. | IDH-wildtype Glioblastoma RTKII | R9 | 35 | IDH-wildtype Glioblastoma RTKII (0.87) | Glioblastoma, IDH-wildtype | *Adult Glioma case* |
| 7. | Choroid Plexus Papiloma type A | R10 | 15 | Plexus tumor, subtype pediatric A (0.90) | Atypical Chorioid Plexus Papiloma | |
| 8. | Inconclusive | R10 | - | Reactive tissue (0.47), Pilocytic Astrocytoma (0.43) | Pilocytic Astrocytoma | *Unusual histological features for pilocytic astrocytoma, histomolecular diagnosis partly based on KIAA1549-BRAF fusion* |
| 9. | Pilocytic Astrocytoma | R10 | 15 | Posterior fossa Pilocytic Astrocytoma (0.99) | Pilocytic Astrocytoma | |
| 10. | IDH-mutant Astrocytoma | R9 | 10 | IDH-mutant, Glioma, High- or Low-Grade Astrocytoma (0.99) | IDH-mutant, Astrocytoma, CNS WHO grade 4 | *Adult Glioma case* |
| 11. | Inconclusive | R10 | - | Not performed | Germinoma | *Germinoma; tumor subtype not in reference Capper et al., 2018 datset. Diagnosis based only on intraoperative frozen section histology* |
| 12. | Inconclusive | R9 | - | White matter (0.99) | IDH-mutant, Astrocytoma, CNS WHO grade 4 | *Adult Glioma case. Sample used for intraoperativesequencing was not representative, Integrated molecular diagnosis based on an other tissue sample.* |
| 13. | Inconclusive | R10 | - | Low-Grade Glioma, *MYB/MYBL1* (0.33) | Diffuse Glioma, Not Elsewhere Classified | *Case considered as 'gliomatosis cerebri' in context of complex genetic background* |
| 14. | Inconclusive | R10 | - | Not performed | Pilocytic Astrocytoma | *Biopsy sample, not tumor representative* |
| 15. | Pilocytic Astrocytoma | R10 | 15 | Pilocytic Astrocytoma subtype: posterior fossa | Pilocytic Astrocytoma | *Resection sample from patient 14* |
| 16. | Adamantinomatous Craniopharyngioma | R10 | 10 | Not performed | Adamantinomatous Craniopharyngioma | *Diagnosis based on histology* |
| 17. | IDH-mutant Astrocytoma | R9 | 45 | Not performed | IDH-mutant, Astrocyto-ma, CNS WHO Grade 2 | *Adult Glioma case* |
| 18. | Pilocytic Astrocytoma | R10 | 40 | Pilocytic Astrocytoma, subtype: posterior fossa (0.95) | Pilocytic Astrocytoma | |
| 19. | Medulloblastoma Group 4 | R10 | 10 | Medulloblastoma, Group 3 and 4, subtype: Group 4 (0.99) | Medulloblastoma, non-*WNT*/non-*SHH* | |
| 20. | Medulloblastoma Group 3 | R10 | 15 | Medulloblastoma, non-*WNT* / non-*SHH*, subtype: Group 3 (0.99) | Medulloblastoma, non-*WNT*/non-*SHH* | |
| 21. | Pilocytic Astrocytoma | R10 | 30 | Pilocytic Astrocytoma (0.97) | Pilocytic Astrocytoma | |
| 22. | Pilocytic Astrocytoma * | R10 | - | Not performed | Pilocytic Astrocytoma | *Recurrent tumor, diagnosis the same as in initial tumor sample* |
| 23. | Ependymoma, subtype *RELA* fusion | R10 | 10 | Ependymoma, subtype *RELA* fusion (0.99) | Supratentorial Ependymoma, *ZFTA* fusion-positive ** | *Classification used to support frozen section diagnosis during surgery* |
| 24. | IDH-wildtype Glioblastoma, subtype Mesenchymal * | R9 | - | IDH-wildtype Glioblastoma, subtype Mesenchymal (0.4) | Diffuse High-Grade Glioma, results further analysis pending | *Adult Glioma case; Imaging consistent with Low-Grade Diffuse Glioma* |
| 25. | Ependymoma, posterior fossa type A | R10 | 25 | Ependymoma, posterior fossa type A (0.99) | Ependymoma, posterior fossa type A | *Classification used to support frozen section diagnosis during surgery* |

**Sturgeon**, indicates the class reached the 0.95 confidence score, asterisk indicates classes that did not reach the 0.95 confidence scores (see Extended Data Fig. 8 and Supplementary Table 8). **CV**, Pore chemistry version. **ST**, Sequencing time until the 0.95 confidence threshold was reached. **Heidelberg**, the diagnosis using the Heidelberg v11b4 classifier and score between parenthesis. **Diagnosis**, lists the integrated histomolecular diagnosis in the 2021 WHO CNS tumor classification system. **Notes**, indicates specific circumstances where applicable. ∗: confidence score >0.95 not reached. ∗∗: RELA fusion positive Ependymoma is renamed to *ZFTA* fusion positive Ependymoma in 2021 WHO CNS tumor classification system.

# Reporting Summary

## Statistics

For all statistical analyses, confirm that the following items are present in the figure legend, table legend, main text, or Methods section.

| n/a | Confirmed | |
|---|---|---|
| ☐ | ☒ | The exact sample size (*n*) for each experimental group/condition, given as a discrete number and unit of measurement |
| ☒ | ☐ | A statement on whether measurements were taken from distinct samples or whether the same sample was measured repeatedly |
| ☒ | ☐ | The statistical test(s) used AND whether they are one- or two-sided<br>*Only common tests should be described solely by name; describe more complex techniques in the Methods section.* |
| ☒ | ☐ | A description of all covariates tested |
| ☒ | ☐ | A description of any assumptions or corrections, such as tests of normality and adjustment for multiple comparisons |
| ☐ | ☒ | A full description of the statistical parameters including central tendency (e.g. means) or other basic estimates (e.g. regression coefficient) AND variation (e.g. standard deviation) or associated estimates of uncertainty (e.g. confidence intervals) |
| ☒ | ☐ | For null hypothesis testing, the test statistic (e.g. *F*, *t*, *r*) with confidence intervals, effect sizes, degrees of freedom and *P* value noted<br>*Give P values as exact values whenever suitable.* |
| ☒ | ☐ | For Bayesian analysis, information on the choice of priors and Markov chain Monte Carlo settings |
| ☒ | ☐ | For hierarchical and complex designs, identification of the appropriate level for tests and full reporting of outcomes |
| ☒ | ☐ | Estimates of effect sizes (e.g. Cohen's *d*, Pearson's *r*), indicating how they were calculated |

*Our web collection on statistics for biologists contains articles on many of the points above.*

## Software and code

Policy information about availability of computer code

| Data collection | MinKNOW v22.10.10, v22.12.7, v22.12.5 |
|---|---|
| Data analysis | ONT Software: Guppy 5.0.11 and v6.3.8, Megalodon 2.5.0., qCat v1.1.0. Code languages: R 4.1.2, Python 3.7.8. R Packages: Rhdf5 2.38.1., QDNAseq v1.30.0, DNAcopy v1.68.0. Python packages: torch v1.10.2, numba v0.56.2, onnxruntime v1.12.1, scipy v1.7.3, scikit-learn v1.0.2, pandas v1.3.5, numpy v1.21.5, matplotlib v3.5.1, pysam v0.19.0, modbampy v0.6.3. Standalone software:  Bedtools v2.30, samtools v1.13 |

For manuscripts utilizing custom algorithms or software that are central to the research but not yet described in published literature, software must be made available to editors and reviewers. We strongly encourage code deposition in a community repository (e.g. GitHub). See the Nature Portfolio guidelines for submitting code & software for further information.

## Data

Policy information about availability of data

All manuscripts must include a data availability statement. This statement should provide the following information, where applicable:
- Accession codes, unique identifiers, or web links for publicly available datasets
- A description of any restrictions on data availability
- For clinical datasets or third party data, please ensure that the statement adheres to our policy

Pediatric patient data is available through the European Genome-Phenome Archive EGAS00001007475, data from AUMC patients is accessible through  NCBI Gene

Expression Omnibus (GEO) with accession code GSE237874. Data used for training is available at GEO, under accession number GSE109381.

## Human research participants

Policy information about [studies involving human research participants and Sex and Gender in Research.](studies involving human research participants and Sex and Gender in Research.)

| Reporting on sex and gender | We refrain from reporting age and sex per individual, as the low number of patients in the Netherlands, and the high diversity of cases could allow identification of individuals based on these parameters. |
|---|---|
| Population characteristics | Data and tissue of the patients of the Princess Máxima Center used in this study were not selected for specific biological criteria. The samples used in this study should therefore represent a cross-section of the overall population of pediatric CNS tumor patients in the Netherlands. Furthermore we do not compare groups of patients and we do not stratify the patient population in any way, and therefore do not see the need to publish these characteristics.<br>However in summary the group of pediatric patients sequenced consists of 35 male and 33 female patients, with age at diagnosis ranging from 0.3 to 16 years (7.2 years on average). These characteristics are available at the repository should other researchers need these for their work. |
| Recruitment | Parents and/or patients (depending on the age) in the Princess Máxima Center signed an informed consent for use of their data and tissue for research purposes (opt-in procedure). For Glioma patients from the Amsterdam University Medical center a broad consent was signed for use of material to improve clinical methods. For AUMC patients we publicly share methylation calls, but not raw DNA sequence data as there is no explicit consent for sharing genetic information. Pediatric patients were included on the basis that they underwent resection surgery for a CNS tumor, and adult patients were selected on the basis that they underwent (suspected) Glioma resection surgery. |
| Ethics oversight | The research was approved by the Biobank and Data access committee (BDAC) of the Princess Máxima Center for pediatric oncology. All included patients provided written informed consent for participation in the biobank (International Clinical Trials Registry Platform: NL7744, https://onderzoekmetmensen.nl/en/trial/21619)). Inclusion of intraoperative samples was ethically approved via the same decision; the results were not shared with caregivers and therefore not used to alter patient treatment nor diagnosis. |

Note that full information on the approval of the study protocol must also be provided in the manuscript.

## Field-specific reporting

Please select the one below that is the best fit for your research. If you are not sure, read the appropriate sections before making your selection.

☒ Life sciences   ☐ Behavioural & social sciences   ☐ Ecological, evolutionary & environmental sciences

For a reference copy of the document with all sections, see [nature.com/documents/nr-reporting-summary-flat.pdf](nature.com/documents/nr-reporting-summary-flat.pdf)

## Life sciences study design

All studies must disclose on these points even when the disclosure is negative.

| Sample size | 94 EPIC profiles, 50 retrospective Nanopore sequencing samples and 25 intraoperative Nanopore sequencing samples are included in this study. Sample sizes were determined based on availability, the 94 methylation array profiles form a representative cross section of patients, and by simulation of shallow sequence runs this number is upsampled<br>to allow extensive model validation.<br>50 retrospective sequencing cases were generated (and again upsampled) to confirm that the findings on array profiles can be translated to real sequencing.<br>25 intraoperative cases is enough to reveal the major challenges in clinical practice, we do not claim this number is enough to calculate sensitivity and specificity; further clinical validation is pending, however the low incidence of CNS tumors and the high number of different classes prevents collection of extensive cohorts in a timely manner. |
|---|---|
| Data exclusions | We did not exclude any samples. |
| Replication | All technical replication attempts were successful, biological replication; ie sequencing from different samples of the same tumor showed some variability, likely due to sample purity; this is adressed in the article. |
| Randomization | Randomization is not applicable to this study: we do not compare groups of patients, but instead compare the results of our method to the existing methylation array and Heidelberg classifier performed for the same patients. |
| Blinding | Blinding was not performed in the study, because the results are based on the classification generated by a neural network algorithm. The black-box nature of such algorithms prevents the operator from influencing the outcome. |

# Reporting for specific materials, systems and methods

We require information from authors about some types of materials, experimental systems and methods used in many studies. Here, indicate whether each material, system or method listed is relevant to your study. If you are not sure if a list item applies to your research, read the appropriate section before selecting a response.

## Materials & experimental systems

| n/a | Involved in the study |
|-----|----------------------|
| ☒ | Antibodies |
| ☒ | Eukaryotic cell lines |
| ☒ | Palaeontology and archaeology |
| ☒ | Animals and other organisms |
| ☐ ☒ | Clinical data |
| ☒ | Dual use research of concern |

## Methods

| n/a | Involved in the study |
|-----|----------------------|
| ☒ | ChIP-seq |
| ☒ | Flow cytometry |
| ☒ | MRI-based neuroimaging |

## Clinical data

Policy information about clinical studies

All manuscripts should comply with the ICMJE guidelines for publication of clinical research and a completed CONSORT checklist must be included with all submissions.

| | |
|---|---|
| Clinical trial registration | This is a biobank study. All included patients provided written informed consent for participation in the biobank (International Clinical Trials Registry Platform: NL7744). |
| Study protocol | Not applicable |
| Data collection | Parents and/or patients (depending on the age) in the Princess Máxima Center signed an informed consent for use of their data and tissue for research purposes (opt-in procedure). The research was approved by the Biobank and Data access committee (BDAC) of the Princess Máxima Center for pediatric oncology. Inclusion of intraoperative samples was ethically approved via the same decision; the results were not shared with caregivers and therefore not used to alter patient treatment nor diagnosis. For Glioma patients from the Amsterdam University Medical center a broad consent was signed for use of material to improve clinical methods. For AUMC patients we publicly share methylation calls, but not raw DNA sequence data as there is no explicit consent for sharing genetic information. |
| Outcomes | Not applicable since this is a biobank study, results were not used to alter treatments. |

