## [Peer Review File · Nature]

Manuscript Title: Ultra-fast deep-learned CNS tumor classification during surgery

Reviewer Comments & Author Rebuttals

Reviewer Reports on the Initial Version:

Referees' comments:

Referee #1 (Remarks to the Author):

The authors describe the feasibility of obtaining diagnostic intraoperative methylation descriptions of CNS brain tumors using nanopore sequencing with a deep learning computational system. They were able to successfully obtain this information within a time frame that will feasibly allow a surgeon to implement diagnostic information into a surgical plan. The results are accurate in the majority of patients. The tumors that created diagnostic confusion are noted, though are rare and not considered tumors that would have significant benefit from intraoperative diagnosis.

Intraoperative nanopore sequencing of CNS tumors is new technology but has been described previously for this patient population:

References

<https://pubmed.ncbi.nlm.nih.gov/28638988/>

<https://www.ncbi.nlm.nih.gov/pmc/articles/PMC8557693/>

<https://pubmed.ncbi.nlm.nih.gov/36269599/>

This burgeoning technology has the potential to create a paradigm shift for intraoperative tumor analysis and extent of surgical resection, influencing not only the efficacy of the procedure but the associated risk to the patient as well. It will be especially critical to have rapid diagnostic information as we continue to move towards molecular classifications of these tumors and individualized precision medicine. Another potential area of benefit will be found in cases where local therapy may be available, and an accurate diagnosis at the time of the original tumor resection may allow these local therapies to be implemented immediately, thus avoiding a second procedure and patient return to the operating room.

The validation process is clearly and coherently described, and the breakdown of clear and difficult diagnoses corresponds to current limitations with more commonplace "traditional" methylation analysis. The optimization of potential unclear results is also addressed in a manner that anticipates the limitations of this method.

Overall, this is a well-written and comprehensive paper describing the implementation of an expedited tissue analysis process that will be important and highly clinically relevant for the management of pediatric brain tumor patients. Pediatric brain tumors are now the most common cause of cancer and the most common cause of medical mortality in children. Improving our ability to care for these patients safely and effectively is of paramount importance in the medical world, and this technology has real and immediate value for children that carry a diagnosis of CNS tumor.

I would suggest the authors expand on the potential clinical implications and limitations of this technology. For instance, is there a minimal tissue volume requirement that may not be met in

situations of small biopsies in critical regions (i.e. DIPG), could this system benefit the (majority of) centers that are unable to obtain molecular diagnosis in a very timely fashion, and thus shorten the time between surgery and implementation of adjuvant therapy thereby improving outcomes for some of the relevant diagnoses, etc.

The references appear to be inclusive of relevant citations. The introduction and conclusions are appropriate.

Referee #2 (Remarks to the Author):

Vermeulen et al described an application of Oxford Nanopore sequencing for CNS tumor classification during surgery. Knowing the cancer subtype can be important for clinical decision making, such as radical resection versus a more conservative strategy. Current standard of care include preoperative imaging and intraoperative histological analysis, but these may not be conclusive or accurate; the current study used molecular profiling through a rapid Nanopore sequencing protocol to assay methylation profiles of tumor samples during surgery, providing an alternative approach to classifying tumor subtype and assisting in intraoperative decision making. Combined with a bioinformatics approach, the authors demonstrated that they can obtain correct diagnosis within 20 to 40 minutes of sequencing data in 45 out of 49 pediatric samples (note that this statement seems to be inconsistent with the experiments performed in Section 2.4 and 2.6). Therefore, the authors demonstrated the high accuracy of intraoperative methylation-based tumor classification and such information may be used to guide decisions intraoperatively (more radical or conservative approach). If the results are generally and broadly applicable to other tumor types for which pre-surgery specimen is hard to obtain for pathology studies, it may have significant implications for treatment decision making and cancer care.

My comments are below, separated by major issues and minor issues:

Major issues:

1. It appears that one major challenge that the Surgeon approach is trying to address is the data sparsity, in that most of the CpG islands are not covered in a typical sequencing run. Indeed, in the first paragraph of Results, they specifically mentioned that to achieve a turn around time (TAT) of 60-90 minutes, only very limited nanopore sequencing data can be generated, in the order of 100-400 Mb. As a result, extremely sparse coverage across the entire genome is expected (covering 0.5-4% of the CpG sites in a 450K array). This suggests potential limitations of the approach compared to for example Illumina 450K arrays, in terms of accuracy of methylation calling, even if the overall patterns of methylation can still be identified by binning genomic regions together. This is a problem that should be further studied in the manuscript, to assess how necessary it is to strictly follow the CpG sites assayed by 450K or EPIC array, when performing their analysis. Alternatively, given that they have substantial number of samples with both array and Nanopore data, they may want to establish equivalence of a collection of nearby CpG sites in the array with a specific CpG region (defined by methylation calls from Nanopore reads on a number of samples, or simply defined in UCSC Genome Browser).

2. Relevant to the previous comment, for the purpose of fast and cheap generation of shallow

sequencing data for methylation analysis, it would be of interest to the readers to see how adaptive sequencing from the Nanopore platform can be used for the same purpose, for a subset of samples. This strategy takes molecules and generates ~500bp sequence data, and it continues sequencing only if the read matches to regions covering known CpG sites annotated in the human genome (otherwise, the read is ejected and the next molecule is taken for sequencing). This is a more direct comparison to Illumina methylation arrays, and previous studies have demonstrated that it has good performance, with results very much comparable to methylation calls from regular ONT whole genome sequencing. It can routinely achieve 5-10X enrichment of methylation sites in the sequencing data in exactly the same amount of sequencing time yet it optimally uses the available time (less than one hour) to generate as much informative data as possible. Since no additional preparation is needed - except a configuration of the sequencing parameters and a BED file listing all CNS tumor relevant methylation regions in the human genome - I would say a typical reader who is familiar with this technology will almost certainly ask the same question as me. It is something worth investigating for even just a small subset of samples, with comparison to data generated without adaptive sequencing in the same amount of time in the same sequencing platform.

3. While methylation profiles have been demonstrated in a few studies to be useful for tumor subtype classification, somatic mutations or alterations can also be predictive of tumor subtypes. The authors alluded to this in the Discussion “While methylation profiles have been demonstrated in a few studies to be useful for tumor subtype classification, somatic mutations or alterations can also be predictive of tumor subtypes.” but did not elaborate on this throughout the manuscript. This is seen as a major comment since large-scale CNVs (such as changes influencing chromosomal arms or >10Mb regions) can be readily seen from shallow sequencing data. The ~300Mb data is essentially 0.1X genome coverage which is far more than sufficient for large-scale or chromosomal level CNVs with very limited time requirement for analysis. The combination of methylation profiles and CNV/CNA profiles may further improve the predictive performance of the machine-learning models and maximally extract available information from sequencing information. The authors have the data already and should make an effort to address this missing piece of information from their prediction models.

4. I am not trying to make things harder for the authors, but as a new manuscript in 2023, it needs to follow the new standards. Megalodon is officially abandoned by the Oxford Nanopore company already (the last release is already one year old and almost certainly will not be maintained further down the road), and nowadays the company suggests Guppy (with remora modification model) for methylation calling after basecalling. Furthermore, R9 flowcell will be discontinued for sale and abandoned by the company within the next a few months as already been announced in 2022. So there are practical considerations that the authors may want to address; to the best of our knowledge, when using R10 flowcells, currently only Guppy and DeepMod2 can provide methylation calls. While many researchers still use R9 flowcells as of today, considering its imminent status of being obsolete in a few months, the authors may want to test a new methylation caller on old R9 data and reproduce the results to show that their method is future-proof (or at least 2024-proof) when users switch to new flowcells. Also just for your reference, check this GitHub issue <https://github.com/nanoporetech/megalodon/issues/328> posted in October 2022 which explains the situation.

5. For the actual machine-learning model, there is no real detail in the manuscript. In section 4.3, I was not able to get a good idea on exactly what types of deep neural network is used and why a deep learning is even needed (compared to say random forest used by the other approaches). What is the dimensionality of the input feature vectors? I can see that the neural network contains three fully connected layers with sigmoid activation but that is really the most I can learn from reading the paper. A justification of the network design (for example, why RELU is not used and instead sigmoid

is used) and how the last layer in the neural network works (for example, whether it is a multi-class classification problem on 91 dimensions with a softmax function for prediction?) is needed.

6. Expensive simulations studies that simulate Nanopore reads are used in the current study. For example, Sturgeon, a deep learning neural network classifier that is patient agnostic, is trained on 14 million and validated on 4 million simulated nanopore runs. The Capper et al. reference dataset with 2801 CNS samples is used for training, validation and testing but this data was generated on arrays, not ONT sequencing. While it has the advantage that the authors can simulate millions of samples, my main concern here is that any types of simulation, such as the one used here, cannot account for the sample-to-sample heterogeneity due to different library qualities and different sequencing characteristics and even the randomness of Nanopore sequencing itself (due to different batches of flowcells produced in different dates). Therefore, while it is promising to see a >0.94 true positive rate within 40 minutes of simulated sequencing, this result should be interpreted with caution and should not be extrapolated to real world settings where much more variability can be introduced into the sequencing run. I think the authors need to significantly tone down the claims for the analysis performed on simulated Nanopore sequencing data throughout the manuscript. It is at most a procedure to train neural network models, and any performance measures on these data sets have no practical implications in real world settings.

Minor issues:

1. The reference style and the organization style for paper is not prepared for Nature.

2. I am not sure what they meant by "Histone 3 mutation". Perhaps it means "Histone H3 mutation" or even "Histone H3.3 mutation"? There are three highly similar histone H3, including H3.1, H3.2, and H3.3, and they are encoded by different genes with high sequence similarity.

3. "nanopore methylation calling has an expected error rate of 10% according to the most performant methylation caller." This statement is untrue. This is obsolete now, and even Nanopore company itself advertises that nanopore methylation calling is more accurate than any other approaches (including array and bisulfite sequencing) and should be considered as the new gold standard for methylation.

4. Can the authors explain why Capper et al. classifier is not used in the study, yet the Heidelberg classifier (V11b4) is used instead? I understand it is an updated version but Capper et al. classifier appears to be more widely known and used. Or perhaps I misunderstood some part of the paper on what is being compared with ONT-based classifier?

5. As mentioned earlier in my major comments, given the concerns on how Nanopore sequencing data simulation reflects the reality encountered during an intraoperative setting, I would suggest the authors to dramatically cut the data simulation part and focus more on the real data analysis (including both retrospective Nanopore analysis of CNS tumors and the four intraoperative cases). Otherwise, the paper loses focus for a typical reader when the more convincing/relevant part of the results are hidden in the last sections of the Results. Also, as I mentioned in the summary paragraph, I am not sure where the "45 out of 49 pediatric samples" number came from since the numbers in Results do not add up together as described.

6. There are a few instances where statement made such as "using a single flowcell". This is confusing and needs to be made more precise, since Nanopore has multiple different types of flowcells with throughput that are orders of magnitude different. Even in this current study they have used both MinION and PromethION.

7. This relates to a major comment that I had above. "This indicated a consistent concordance of 88-90% between binarized array data (beta cutoff at >0.6) and nanopore methylation calls" is a little concerning, since for all the data that we had access to, we can see a much higher consistency between Illumina 450K array and Nanopore sequencing based methylation calls on the same CpG site, typically over 0.95. The ONT company also had similar observations as shown in their white paper and posters. (Note that the authors did not even use the same CpG site, they used a 50 base pairs windows centered on the CpG site targeted by each Infinium probe for comparisons which probably means the value can be even lower when using the same CpG site.) I think this needs further investigation and a different methylation caller can help.

Referee #3 (Remarks to the Author):

In the manuscript "Ultra-fast deep-learned pediatric CNS tumor classification during surgery" Vermeulen et al., describe an innovative classifier based on neural networks (NN) to categorize brain tumours, intraoperatively, into sub-classes based on methylation profiles obtained rapidly from nanopore devices, rather than existing ad-hoc random-forest classifiers that must be custom built for each patient at run-time. They demonstrate the feasibility of tumor classification within a turn-around time of an hour-and-half, and they describe the benefits of a NN -based approach. In addition to speed and generalizability, the model can be distributed between institutes and updated centrally while preserving patient privacy, due to the obfuscation of patient data provided by the NN. Owing to the paucity of existing nanopore training data for the many tumor types, Illumina DNA methylation array-based data is used as a proxy with which to develop synthetic nanopore "reads" to be used as a training basis.

This work represents a natural and important progression in the field of DNA methylation based tumor classification. The significance of the work is aptly and correctly summarized by the authors: "Sturgeon uniquely moves the computationally intensive model training, validation and calibration phase outside the surgical time window", The introduction provides a well-researched introduction to the field and establishes the necessary context and motivation for the work. That being said, I have several concerns about the presentation of the material, and offer the following observations:

Major points

- Page 3: The NN model architecture is not clear. In Sec. 4.3 it is noted that there are 3 fully connected layers of length 256, 128, and [Nclasses] respectively. This defines the "output" end clearly, but what about the input? is the first layer of 256 nodes fully connected to a previous layer with a node for each of the thousands of probe sites? Where exactly is the binarised input received? The opportunity to review the authors code would help to resolve this ambiguity, but for the general reader, a figure with the NN architecture would be very helpful (and might be better prioritized above content currently used in other figures).

- On page 4: the manuscript states "to account for the fact that in the sparse setting, where the maximum coverage is 1x..." While coverage exceeding 1x may be rare, the phrasing here suggests it is a theoretical maximum; there is however, no such constraint in principle. As the authors note the potential for >1x coverage in Sec. 4.13 ("When multiple reads cover the same Infinium probe site, majority voting is also used..."). Likewise, in section 4.1, "Short nanopore sequencing runs yield low (<1X)" should read " $\leq 1 X$ ". Is the potential for overlapping reads reflected in the randomized start positions of the simulated reads? What is the procedure when voting results in a tie?

- The authors describe the generation synthetic nanopore "reads". In a real nanopore run, coverage

probability is non-uniformly distributed, as certain genomic regions are less accessible, and in case of cancer, chromosomal regions may be gained/amplified or present as monozygous state or even deleted and this will have strong impact on the nanopore read distribution. Please comment how this was addressed and by the authors and how and how non-uniform coverage might affect the model's performance.

- The use of Illumina DNA methylation array data as a proxy for simulated nanopore reads is an important and innovative solution. A figure quantifying the agreement (or discrepancy) between these two technologies would be helpful in validating this step.
- In Section 2.4 the manuscript states "To assess the performance of Sturgeon in a realistic setting...". To me the setting that follows is not a realistic intrasurgical setting but merely an analysis of retrospective frozen samples. This should be rephrased.
- The frozen test samples only cover relatively few of the classes that are available in the classifier. This should be expanded as it is not clear whether performance generalizes to all classes. The authors should also consider to add adult cases.
- Sturgeon is intended to be applied in an intraoperative setting. In such a setting tumor cell content cannot be controlled as well as in formalin fixed paraffin embedded samples. Indeed, two of the four samples with true intraoperative analysis had technical difficulties with no or few tumor cells. Therefore, it seems highly relevant to test the robustness, validity and reliability of Sturgeon in a larger cohort of intraoperative samples including low tumor cell samples. This should be done by 1) analyzing a far larger number of true intraoperative samples to observe how often low tumor content phenomenon arises and how Sturgeon classifies such samples 2) analyzing a number of cases with histologically proven low tumor cell content and determining the performance of Sturgeon for those. In addition, the methodology employed in the manuscript for generating synthetic nanopore reads could be extended to create *in silico* mixtures of tumors with precisely defined ratios, in order to probe the sensitivity of the model to this phenomenon. Single-read-level analysis might also offer a solution to cell-type mixture deconvolution in realistic data.
- On page 12, the manuscript states: "We find that Sturgeon outperforms nanoDx, the patient-specific random forest classifier, as it is able to correctly predict 9 additional samples (Supplementary figure 12)"; However, from looking at supplementary Figure 12, it is not clear how this conclusion should be reached --i.e. which 9 additional samples are being referred to, and how Sturgeon outperforms nanoDx. Supp. Fig. 12 seems to show 27 cases where nanoDx is correct where Sturgeon is incorrect, compared to only 26 cases vice-versa (i.e. nanoDx slightly outperforming Sturgeon). Please clarify.

Minor points

- Font labels for panels ("a", "b", "c") are inconsistent between figures.
- Fig 2d It is unclear what the "*" symbol signifies.
- Fig. 3 might be condensed, or moved to a supplement.
- The caption in Figure 4 states "Asterisks indicate the first time point where the score of the correct class was higher than 0.8." The asterisks in the figure actually appear to correspond to the point where the threshold 0.95 is crossed, not 0.8 as stated in the caption.

- In Fig 4a, 5a - the x-axis marks are overly dense and non-uniform, making it difficult to draw comparison between adjacent plots. Fewer, evenly spaced ticks at shared positions (e.g. 5000, 10,000, 15,000) would ease readability.
- In Fig 4a,
 - the abbreviations may not be known to the average reader.
 - A few illustrative examples should be employed, with the remainder moved to the supplement.
- In Fig 4c
 - different symbols (diamonds/squares/shapes, etc.) should be used instead relying only on colours, as the orange and red are difficult to distinguish for colour-blind readers
 - The legend should clarify that colour codes denote when "only NanoDx" is "correct", likewise "only Sturgeon";
- On Page 12: "For each sample we generated 200.000 reads (3.9Gb)" the abbreviation for gigabases should be stated explicitly, to avoid confusion with "Gigabytes" (of data)
- In Fig. 5a, the caption states "only correct classification classes are displayed", with upward-sloping circle data points corresponding to the two models; it is unclear what the downward-sloping diamond symbols represent. Please recheck this figure.
- Figure 19 from the supplement is missing. Only 18 figures are shown.
- The supplementary video on YouTube has no audio; perhaps it was redacted for the privacy of laboratory staff, but a note of clarification to that effect should be added.
- Word economy: multiple long sentences are occasionally used where fewer, shorter sentences could convey the same information more clearly. The abstract in particular could be condensed.
- It is appropriate that the bulk of the paper is written to ease reading, while more detailed discussion is relegated to, e.g. methods. However, for a more full picture, references to detailed sections should be provided where an overview leaves room for uncertainty. More sentences like "as described in section x/y ..." Are necessary. For example:
 - page 12- subsampling methodology, ( "refer to methods"....)
 - The definition of "correct" vs. "incorrect" classifier predictions require clearly defined ground truth not established until section 4.9.
 - Page 3: "Our final models are trained on 14 million and validated on 4 million simulated nanopore runs," It is not yet clear at this point what a "run" represents; the reader should be referred to sec. 4.7 (page 23) where this is clarified.
- On Page 13: it is written "We note that practically all of the misdiagnoses arise from two of the 35 samples". This qualitative statement should be made quantitative : e.g. "We note that x of y misdiagnoses..."
- The F1-score metric should be defined when introduced for the benefit of general readers.
- On Page 4 the manuscript states: "Sturgeon therefore employs a data augmentation approach to effectively upsample the number of training samples available. This approach also allows for class-balancing by upsampling small classes relatively more compared to larger classes."
 - Here, the 2nd usage of "upsampling" is a clear reference to balancing rare-vs-common class types. However, the first usage of "upsampling" is unclear from the context and should be clarified.

Author Rebuttals to Initial Comments:

Referee #1 (Remarks to the Author):

The authors describe the feasibility of obtaining diagnostic intraoperative methylation descriptions of CNS brain tumors using nanopore sequencing with a deep learning computational system. They were able to successfully obtain this information within a time frame that will feasibly allow a surgeon to implement diagnostic information into a surgical plan. The results are accurate in the majority of patients. The tumors that created diagnostic confusion are noted, though are rare and not considered tumors that would have significant benefit from intraoperative diagnosis.

Intraoperative nanopore sequencing of CNS tumors is new technology but has been described previously for this patient population:

References

<https://pubmed.ncbi.nlm.nih.gov/28638988/>

<https://www.ncbi.nlm.nih.gov/pmc/articles/PMC8557693/>

<https://pubmed.ncbi.nlm.nih.gov/36269599/>

This burgeoning technology has the potential to create a paradigm shift for intraoperative tumor analysis and extent of surgical resection, influencing not only the efficacy of the procedure but the associated risk to the patient as well. It will be especially critical to have rapid diagnostic information as we continue to move towards molecular classifications of these tumors and individualized precision medicine. Another potential area of benefit will be found in cases where local therapy may be available, and an accurate diagnosis at the time of the original tumor resection may allow these local therapies to be implemented immediately, thus avoiding a second procedure and patient return to the operating room.

The validation process is clearly and coherently described, and the breakdown of clear and difficult diagnoses corresponds to current limitations with more commonplace “traditional” methylation analysis. The optimization of potential unclear results is also addressed in a manner that anticipates the limitations of this method.

Overall, this is a well-written and comprehensive paper describing the implementation of an expedited tissue analysis process that will be important and highly clinically relevant for the management of pediatric brain tumor patients. Pediatric brain tumors are now the most common cause of cancer and the most common cause of medical mortality in children. Improving our ability to care for these patients safely and effectively is of paramount importance in the medical world, and this technology has real and immediate value for children that carry a diagnosis of CNS tumor.

COMMENT

I would suggest the authors expand on the potential clinical implications and limitations of this technology. For instance, is there a minimal tissue volume requirement that may not be met in situations of small biopsies in critical regions (i.e. DIPG), could this system benefit the (majority of) centers that are unable to obtain molecular diagnosis in a very timely fashion, and thus shorten the time between surgery and implementation of adjuvant therapy thereby improving outcomes for some of the relevant diagnoses, etc.

We thank the reviewer for the constructive remarks and suggestions. We have now included more discussion on the limitations and potential future applications in the discussion.

In regards to the sample limitations we currently ask for relatively small (5x5x5mm) samples, which already yields micrograms of DNA, while only 200 ng is needed. Thus the DNA extraction protocol will likely work for even smaller samples as well, and even for very small samples if a DNA concentration step is included. We have performed sequencing on several Diffuse Intrinsic Pons Glioma (currently referred to as Glioblastoma H3K27 altered) samples where DNA yield was sufficient for nanopore sequencing.

Changes to manuscript:

- We discuss the potential sample mass limitations in the discussion on line 629-636.
- We discuss the applicability in peripheral centers with limited financial resources on lines 610-621.
- We reference the potential future application of Sturgeon in intraoperative drug delivery on lines 622-628.

The references appear to be inclusive of relevant citations. The introduction and conclusions are appropriate.

Referee #2 (Remarks to the Author):

Vermeulen et al described an application of Oxford Nanopore sequencing for CNS tumor classification during surgery. Knowing the cancer subtype can be important for clinical decision making, such as radical resection versus a more conservative strategy. Current standard of care include preoperative imaging and intraoperative histological analysis, but these may not be conclusive or accurate; the current study used molecular profiling through a rapid Nanopore sequencing protocol to assay methylation profiles of tumor samples during surgery, providing an alternative approach to classifying tumor subtype and assisting

in intraoperative decision making. Combined with a bioinformatics approach, the authors demonstrated that they can obtain correct diagnosis within 20 to 40 minutes of sequencing data in 45 out of 49 pediatric samples (note that this statement seems to be inconsistent with the experiments performed in Section 2.4 and 2.6). Therefore, the authors demonstrated the high accuracy of intraoperative methylation-based tumor classification and such information may be used to guide decisions intraoperatively (more radical or conservative approach). If the results are generally and broadly applicable to other tumor types for which pre-surgery specimen is hard to obtain for pathology studies, it may have significant implications for treatment decision making and cancer care.

My comments are below, separated by major issues and minor issues:

Major issues:

1. It appears that one major challenge that the Sturgeon approach is trying to address is the data sparsity, in that most of the CpG islands are not covered in a typical sequencing run. Indeed, in the first paragraph of Results, they specifically mentioned that to achieve a turn around time (TAT) of 60-90 minutes, only very limited nanopore sequencing data can be generated, in the order of 100-400 Mb. As a result, extremely sparse coverage across the entire genome is expected (covering 0.5-4% of the CpG sites in a 450K array). This suggests potential limitations of the approach compared to for example Illumina 450K arrays, in terms of accuracy of methylation calling, even if the overall patterns of methylation can still be identified by binning genomic regions together. This is a problem that should be further studied in the manuscript, to assess how necessary it is to strictly follow the CpG sites assayed by 450K or EPIC array, when performing their analysis. Alternatively, given that they have substantial number of samples with both array and Nanopore data, they may want to establish equivalence of a collection of nearby CpG sites in the array with a specific CpG region (defined by methylation calls from Nanopore reads on a number of samples, or simply defined in UCSC Genome Browser).

We thank the reviewer for the careful analysis of our manuscript and the insightful comments and suggestions. We agree with the reviewer that nearby CpG sites are very likely to have an identical methylation state. In the original analysis we use a 50bp window centered on each CpG site in the 450K methylation array to take a majority vote for CpG methylation state over multiple sites. We found that this already corrects methylation calling inaccuracies. To further demonstrate the effect of this window size we now also include comparisons between methylation array data and nanopore data at different window sizes and show that a marginally (<0.5%) higher similarity is obtained at slightly larger (100bp) window sizes (for R9 chemistry) and that also more sites are interpretable. Based on these analyses, we adjusted our pipeline and now use a 100bp window for the R9 chemistry data. For R10 chemistry, with an innately higher methylation calling accuracy, we observed that

the methylation accuracy is already very high when only taking the CpG site itself into account, but we chose to maintain the 100bp window for simplicity as it does not show a drop in performance.

Changes to manuscript:

- Supplementary table 1 was added to further elaborate on the use of different sizes of methylation calling windows.
- Supplementary figure 1 was added to further elaborate on the relation between EPIC scores and nanopore sequencing based methylation calls.
- Re-processed all the sequenced samples (retrospective and intraoperative) with the 100bp window.
- The main text was edited to better explain the reason for discrepant EPIC vs (shallow) nanopore sequencing calls (lines 161-167).

2. Relevant to the previous comment, for the purpose of fast and cheap generation of shallow sequencing data for methylation analysis, it would be of interest to the readers to see how adaptive sequencing from the Nanopore platform can be used for the same purpose, for a subset of samples. This strategy takes molecules and generates ~500bp sequence data, and it continues sequencing only if the read matches to regions covering known CpG sites annotated in the human genome (otherwise, the read is ejected and the next molecule is taken for sequencing). This is a more direct comparison to Illumina methylation arrays, and previous studies have demonstrated that it has good performance, with results very much comparable to methylation calls from regular ONT whole genome sequencing. It can routinely achieve 5-10X enrichment of methylation sites in the sequencing data in exactly the same amount of sequencing time yet it optimally uses the available time (less than one hour) to generate as much informative data as possible. Since no additional preparation is needed - except a configuration of the sequencing parameters and a BED file listing all CNS tumor relevant methylation regions in the human genome - I would say a typical reader who is familiar with this technology will almost certainly ask the same question as me. It is something worth investigating for even just a small subset of samples, with comparison to data generated without adaptive sequencing in the same amount of time in the same sequencing platform.

We thank the reviewer for this suggested improvement. Originally we did not use adaptive sampling, since the initial samples in our cohort were sequenced before adaptive sampling became widely available and we preferred to keep the protocol consistent and simple. We agree with the reviewer that adaptive sampling is a straightforward way to obtain more informative reads per minute, and that informing the readers of this opportunity is important and ensures our manuscript can inspire diagnostic labs to make use of this option. We have therefore re-sequenced five samples with adaptive sampling enabled on half of the

channels to quantify the improvement this method offers compared to regular nanopore sequencing. We observe that the informative CpG sites sequenced per minute is increased by approximately 15-30%. It should be noted, however, that we encountered several technical challenges in getting adaptive sampling to work (related to GPU utilization on our workstation), and in the end only succeeded in running it on a GridION device (details of the setup described in the Method/Supplementary methods section). Using the GridION device, which is much more costly and requires a dedicated lab setup, may be possible for some diagnostic labs, but there are practical advantages of a MinION tailored system which is why we used MinION (non-adaptive sampling) for intraoperative cases.

Changes to manuscript:

- We added a paragraph about adaptive sampling to the Results section (Adaptive sampling can further reduce sequencing time): lines 521-548
- We added adaptive sampling results to figure 5, and description,
- We added supplementary figure 30 and 31 to show the performance,
- We added a methods section to explain how we set up adaptive sampling (lines 949-966).

3. While methylation profiles have been demonstrated in a few studies to be useful for tumor subtype classification, somatic mutations or alterations can also be predictive of tumor subtypes. The authors alluded to this in the Discussion “While methylation profiles have been demonstrated in a few studies to be useful for tumor subtype classification, somatic mutations or alterations can also be predictive of tumor subtypes.” but did not elaborate on this throughout the manuscript. This is seen as a major comment since large-scale CNVs (such as changes influencing chromosomal arms or >10Mb regions) can be readily seen from shallow sequencing data. The ~300Mb data is essentially 0.1X genome coverage which is far more than sufficient for large-scale or chromosomal level CNVs with very limited time requirement for analysis. The combination of methylation profiles and CNV/CNA profiles may further improve the predictive performance of the machine-learning models and maximally extract available information from sequencing information. The authors have the data already and should make an effort to address this missing piece of information from their prediction models.

The reviewer makes a good point that Copy Number Variations (CNV) can be important to further improve diagnostic accuracy and that shallow nanopore sequencing may provide sufficient coverage to yield a (noisy) CNV profile with diagnostic utility. Following up on this, we confirmed for 6 cases that a very reasonable CNV profile can be obtained using only 20,000-50,000 sequence reads (corresponding to approximately 30-40 minutes of sequencing). While limited to large scale CNVs, pathologists can interpret these profiles in a similar fashion to microarray-based profiles and take them into account as supporting

evidence when deciding on a final diagnosis. We have adapted our standard workflow (please see the reference Github page) to include CNV plots as the standard output.

The reviewer furthermore alludes to the possibility of including CNV profiles in the Machine Learning model. While this may certainly be a good idea, we feel this is outside the scope of methylation-based classification and moreover comes with clear drawbacks, such as the need for integration of different types of features (requiring normalization and feature balancing) and increased risk of overfitting. The latter is caused by the fact that CNV profiles at this shallow depth can only point to large (chromosome-level) CNAs and therefore contain very limited variability so that the extensive data augmentation performed in Sturgeon will not yield sufficiently diverse training data and may overtrain on specific (potentially spurious or irrelevant) CNAs present in a single sample.

Finally, we also tested the robustness of our classifier to samples that exhibit non-uniform CNV profiles. For this, we changed our simulation to take into account specific CNV profiles, by sampling sequencing reads according to a probability dependent on the genomic position. We then compared the results of our 'normal' cross-validation simulation, with the non-uniformly simulated 40 minute sequencing runs. Four different non-uniform profiles were simulated:

- Two clinically relevant profiles: which we call glioblastoma (Chr7 amplification and Chr10 deletion) and oligodendroglioma (Chr1 and Chr19 deletions) due to their prevalence on these CNS types.
- Two extreme cases to test for robustness: which we call odd (all even chromosomes deleted) and even (all odd chromosomes deleted).

Our results show that the model is extremely robust. The observed changes in F1-scores are minute for any of the non-uniform CNV profiles compared to the uniform CNV profile simulations that were used for training. Together with all the nanopore sequencing runs on real samples (which also exhibit non-uniform CNV profiles), this clearly demonstrates that Sturgeon is very robust to the sampling due to copy-number changes.

CNV simulation analysis. Sturgeon performance (F1-score) simulations test fold samples from the reference dataset from Capper et al., 2018. We compared uniform genomic sampling ($N=5,000,000$) against: extreme sampling (only odd or even chromosomes) and biologically relevant CNV profiles for glioblastoma (Chr7 amplification and Chr10 deletion) and oligodendroglioma (Chr1 and Chr19 deletions), ($N=500,000$ for each CNV profile). We evaluated all four Sturgeon submodels on the test fold samples, each sample was simulated 50 times using test fold seeds.

Changes to manuscript:

- We added a paragraph about CNV profiles (lines 357-385),
- We added a description of CNV calling to the methods (lines 917-929).
- We added examples to figure 3, and supplementary figures 19-21.

4. I am not trying to make things harder for the authors, but as a new manuscript in 2023, it needs to follow the new standards. Megalodon is officially abandoned by the Oxford Nanopore company already (the last release is already one year old and almost certainly will not be maintained further down the road), and nowadays the company suggests Guppy (with remora modification model) for methylation calling after basecalling. Furthermore, R9 flowcell will be discontinued for sale and abandoned by the company within the next a few months as already been announced in 2022. So there are practical considerations that the authors may want to address; to the best of our knowledge, when using R10 flowcells, currently only Guppy and DeepMod2 can provide methylation calls. While many researchers still use R9 flowcells as of today, considering its imminent status of being obsolete in a few months, the authors may want to test a new methylation caller on old R9 data and reproduce the results to show that their method is future-proof (or at least 2024-proof) when users switch to new flowcells. Also just for your reference, check this GitHub issue <https://github.com/nanoporetech/megalodon/issues/328> posted in October 2022 which explains the situation.

We thank the reviewer for sharing their expertise on this matter. We are aware of the deprecation of Megalodon, and also slightly disappointed by this decision. Our decision to use Megalodon is based on an in depth benchmarking (<https://doi.org/10.1038/s41467-021-23778-6>), which clearly showed that Megalodon is the most performant “out-of-the-box” methylation calling tool in terms of accuracy and recall. This is important, as in the sparse setting, every call counts. Indeed in our hands, and for our application, Megalodon with a Rerio model outperforms Megalodon with a Remora model for R9 flowcells.

But we certainly agree with the reviewer that our work will be most valuable if we can show the results are robust to the choice of the caller and future sequencing flowcells (assuming future releases are at least equivalent in terms of accuracy).

To demonstrate robustness and show that Sturgeon is ‘future proof’, we re-sequenced 5 samples on R10 flowcells, show the similarity to EPIC profiles using Remora and demonstrate classification performance remains very high. Furthermore, we migrated to R10 flowcells for new intraoperative cases. Of note: we observed that the R10 library prep workflow ‘out of the box’ yields a different read length distribution (a more dispersed range of fragment sizes) compared to R9 chemistry, and include an optimised R10 workflow in the methods section which will aid adoption of Sturgeon by the community.

Changes to manuscript:

- We added benchmarking results for the R10 sequenced samples to supplementary table 1.
- We processed intraoperative samples with R10 and included the specific chemistry version in Table 1.
- We added a comparison between R9 and R10 chemistry in supplementary figure 22.

- We added a methods description for our R10 workflow for the library prep, flowcell loading and methylation calling sections.

5. For the actual machine-learning model, there is no real detail in the manuscript. In section 4.3, I was not able to get a good idea on exactly what types of deep neural network is used and why a deep learning is even needed (compared to say random forest used by the other approaches). What is the dimensionality of the input feature vectors? I can see that the neural network contains three fully connected layers with sigmoid activation but that is really the most I can learn from reading the paper. A justification of the network design (for example, why RELU is not used and instead sigmoid is used) and how the last layer in the neural network works (for example, whether it is a multi-class classification problem on 91 dimensions with a softmax function for prediction?) is needed.

We agree with the reviewer that more detail around the machine-learning is appropriate. We have therefore rewritten the description of the machine-learning model to improve clarity around the design choices of the architecture, parameter settings and optimization. Furthermore, we include a Supplementary Figure that should make it easier to understand the architecture of the network.

The motivation for a deep neural network (DNN) is two-fold: first of all, the simulation upsampling approach generates vast amounts of training data (which is required to deal with the sparsity-noise). This is naturally dealt with by the batch-wise training of DNNs, and one of the reasons why such models have been so successful in domains where millions or more noisy training examples are available. A random-forest (RF) approach, for instance, would need to be trained on all data at once which is computationally infeasible. Alternatively, (very) many RFs could be trained on small batches, but then computation time during inference would increase. Secondly, DNNs can be iteratively improved when new data becomes available without requiring access to all previous training data. This is a major advantage as this in theory allows centres to refine the model to their own patient data.

Changes to manuscript:

- We added Supplementary figure 2 for a graphic summary of the ML method. With clear input and output dimensions.
- We expanded the methods description “Neural network architecture” to elaborate on the machine learning details. (lines 693-705)
- Added a section in the discussion to explain the potential for federated learning across different institutes (lines 592-603).

6. Expensive simulations studies that simulate Nanopore reads are used in the current study. For example, Sturgeon, a deep learning neural network classifier that is patient agnostic, is

trained on 14 million and validated on 4 million simulated nanopore runs. The Capper et al. reference dataset with 2801 CNS samples is used for training, validation and testing but this data was generated on arrays, not ONT sequencing. While it has the advantage that the authors can simulate millions of samples, my main concern here is that any types of simulation, such as the one used here, cannot account for the sample-to-sample heterogeneity due to different library qualities and different sequencing characteristics and even the randomness of Nanopore sequencing itself (due to different batches of flowcells produced in different dates). Therefore, while it is promising to see a >0.94 true positive rate within 40 minutes of simulated sequencing, this result should be interpreted with caution and should not be extrapolated to real world settings where much more variability can be introduced into the sequencing run. I think the authors need to significantly tone down the claims for the analysis performed on simulated Nanopore sequencing data throughout the manuscript. It is at most a procedure to train neural network models, and any performance measures on these data sets have no practical implications in real world settings.

We agree with the reviewer that simulated nanopore runs are not equivalent to actual nanopore runs. We included these analyses to offer insight into how the classifier works and – most importantly – study its limits. The true performance can only be established through validation on nanopore sequencing. For this reason, and also in response to the comments of reviewer nr 3, we now put less emphasis on the simulation results, and more on the performance on nanopore sequencing data and intraoperative cases. We would also like to point out that the manuscript includes demonstration of the efficacy of the Sturgeon classifier on >400 samples sequenced by Kuschel *et al.*, which were publicly available (figure 3). This dataset consists of samples sequenced in another institute, methylation called using a different pipeline, and data in an already processed format where only the methylation status of probe sites is available. Despite these differences, Sturgeon still performs on par with our own samples and similar to the simulations, demonstrating its robustness. We have reordered the presentation of the results to emphasize this.

Changes to manuscript:

- We shortened the description of cross-validation and on retrospective validation on pediatric 450K profiles and moved some details to the accompanying Supplementary Table 3.
- We condensed Figure 2 (previously Figure 3) and added an additional analysis on sample purity (lines 269-300) .

Minor issues:

1. The reference style and the organization style for paper is not prepared for Nature.

We adjusted the reference style and organization accordingly.

2. I am not sure what they meant by “Histone 3 mutation”. Perhaps it means “Histone H3 mutation” or even “Histone H3.3 mutation”? There are three highly similar histone H3, including H3.1, H3.2, and H3.3, and they are encoded by different genes with high sequence similarity.

Rephrased to the appropriate term: Glioblastoma, subtype H3K27 altered, which is used to indicate a Histone 3 gene, mutated at the lysine residue at position 28 (p.K28) which defines this distinct tumor entity. Unfortunately, the original papers referred to this mutation as p.K27 (now known to be incorrect), but since H3K27 has been used for many years, the name stuck and is still being used in literature.

3. “nanopore methylation calling has an expected error rate of 10% according to the most performant methylation caller.” This statement is untrue. This is obsolete now, and even Nanopore company itself advertises that nanopore methylation calling is more accurate than any other approaches (including array and bisulfite sequencing) and should be considered as the new gold standard for methylation.

We would like to clarify (and have updated the manuscript to reflect this clarification) that our use case, and therefore method of benchmarking, is different to those used by others (published or otherwise). Benchmarks are generally performed on homogeneous cell populations, preferably limited to CpG sites with a clear methylated or unmethylated state or using a methylated fraction per site for comparison between methods. In our application, however, we expect impure and possibly heterogeneous samples.

The 10% error rate we mention in the manuscript is a combination of multiple factors including methylation calling accuracy, but upon further analysis we discovered that it mostly results from heterogeneously methylated sites that are binarized from the methylation arrays. We added an analysis where we adjust the binarization of methylation profiles to be more stringent, and indeed when we only analyse clearly methylated or unmethylated sites, the accuracy is >90% (Supplementary figure x, copied below as well). While using a stricter cutoff would increase the correlation between EPIC methylation profiling and nanopore sequencing, it comes at the cost of recall, and thus we prefer to use a more lenient approach at the cost of some accuracy (also explained in the manuscript).

See supplementary figure 1; when we use a single cutoff to binarize Infinium profiles, the maximum concordance with nanopore sequencing is <95%. When we use a two-sided cutoff, discarding all the beta values that fall in the middle, we can obtain a much higher concordance, at the cost of the number of usable sites. From this we conclude that the major source of errors is not necessarily methylation calling, but also binarizing heterogeneous sites in Infinium arrays and nanopore sequencing them with low coverage.

Changes to manuscript:

- We added Supplementary figure 1 with more EPIC vs nanopore sequencing comparisons (shown above).
- We rephrased our statement regarding the expected discrepancy between nanopore methylation calling and Infinium arrays (lines 161-167).

4. Can the authors explain why Capper et al. classifier is not used in the study, yet the Heidelberg classifier (V11b4) is used instead? I understand it is an updated version but Capper et al. classifier appears to be more widely known and used. Or perhaps I misunderstood some part of the paper on what is being compared with ONT-based classifier?

The Heidelberg classifier is a publicly available adaptation of the Capper classifier, available via the DKFZ website and used in clinical practice.

In the manuscript (line 206-208) this is now stated as:

“This classifier can be considered an updated version of the Capper et al. classifier since it is based on an extremely comprehensive collection of EPIC profiles.”

5. As mentioned earlier in my major comments, given the concerns on how Nanopore sequencing data simulation reflects the reality encountered during an intraoperative setting, I would suggest the authors to dramatically cut the data simulation part and focus more on the real data analysis (including both retrospective Nanopore analysis of CNS tumors and the four intraoperative cases). Otherwise, the paper loses focus for a typical reader when the more convincing/relevant part of the results are hidden in the last sections of the Results. Also, as I mentioned in the summary paragraph, I am not sure where the “45 out of 49 pediatric samples” number came from since the numbers in Results do not add up together as described.

We have adjusted the emphasis more towards nanopore sequenced samples, and we moved the intraoperative results to an earlier point in the manuscript.

6. There are a few instances where statement made such as “using a single flowcell”. This is confusing and needs to be made more precise, since Nanopore has multiple different types of flowcells with throughput that are orders of magnitude different. Even in this current study they have used both MinION and PromethION.

We now specify the type of flowcells throughout the manuscript.

7. This relates to a major comment that I had above. “This indicated a consistent concordance of 88-90% between binarized array data (beta cutoff at >0.6) and nanopore methylation calls” is a little concerning, since for all the data that we had access to, we can see a much higher consistency between Illumina 450K array and Nanopore sequencing based methylation calls on the same CpG site, typically over 0.95. The ONT company also had similar observations as shown in their white paper and posters. (Note that the authors did not even use the same CpG site, they used a 50 base pairs windows centered on the CpG site targeted by each Infinium probe for comparisons which probably means the value can be even lower when using the same CpG site.) I think this needs further investigation and a different methylation caller can help.

We thank the reviewer for raising this point. As per the suggestion by the reviewer, we now include additional benchmarking with a more recent version of Guppy. However, as described in our reply to minor issue #3, we do not believe a 95% accuracy can be achieved

when comparing binarized methylation data to nanopore sequencing, for the simple fact alone that some sites will be heterogeneously methylated in different alleles and/or cells. We also would like to mention that we will make all the raw data available for future users to test their methylation callers and potentially improve performance.

Referee #3 (Remarks to the Author):

In the manuscript "Ultra-fast deep-learned pediatric CNS tumor classification during surgery" Vermeulen et al., describe an innovative classifier based on neural networks (NN) to categorize brain tumours, intraoperatively, into sub-classes based on methylation profiles obtained rapidly from nanopore devices, rather than existing ad-hoc random-forest classifiers that must be custom built for each patient at run-time. They demonstrate the feasibility of tumor classification within a turn-around time of an hour-and-half, and they describe the benefits of a NN -based approach. In addition to speed and generalizability, the model can be distributed between institutes and updated centrally while preserving patient privacy, due to the obfuscation of patient data provided by the NN. Owing to the paucity of existing nanopore training data for the many tumor types, Illumina DNA methylation array-based data is used as a proxy with which to develop synthetic nanopore "reads" to be used as a training basis.

This work represents a natural and important progression in the field of DNA methylation based tumor classification. The significance of the work is aptly and correctly summarized by the authors: "Sturgeon uniquely moves the computationally intensive model training, validation and calibration phase outside the surgical time window", The introduction provides a well-researched introduction to the field and establishes the necessary context and motivation for the work. That being said, I have several concerns about the presentation of the material, and offer the following observations:

Major points

- Page 3: The NN model architecture is not clear. In Sec. 4.3 it is noted that there are 3 fully connected layers of length 256, 128, and [Nclasses] respectively. This defines the "output" end clearly, but what about the input? is the first layer of 256 nodes fully connected to a previous layer with a node for each of the thousands of probe sites? Where exactly is the binarised input received? The opportunity to review the authors code would help to resolve this ambiguity, but for the general reader, a figure with the NN architecture would be very helpful (and might be better prioritized above content currently used in other figures).

We agree with the reviewer that more explanation of the neural network would be helpful, and now add a supplementary figure and more extensive explanation to clarify the architecture. Please also see our response to a similar point raised by Reviewer 2.

Changes to manuscript:

- We added Supplementary figure 2 for a graphic summary of the ML method. With clear input and output dimensions.
 - We expanded the methods description "Neural network architecture" to elaborate on the machine learning details.
- On page 4: the manuscript states "to account for the fact that in the sparse setting, where the maximum coverage is 1x..." While coverage exceeding 1x may be rare, the phrasing here suggests it is a theoretical maximum; there is however, no such constraint in principle. As the authors note the potential for >1x coverage in Sec. 4.13 ("When multiple reads cover the same Infinium probe site, majority voting is also used..."). Likewise, in section 4.1, "Short nanopore sequencing runs yield low (<1X)" should read "($\leq 1 X$)". Is the potential for overlapping reads reflected in the randomized start positions of the simulated reads? What is the procedure when voting results in a tie?

We thank the reviewer for this comment and apologise for any confusion we may have caused. Indeed, we should have phrased this differently, and we now adjusted the text accordingly: we expect coverage to be 0 or 1x for the majority of sites, with some exceptions having higher coverage. In case of a tie we disregard the site altogether

This situation does not play a role for simulated reads, as these have identical methylation state and are thus always in agreement.

-We adjusted the text (line 154-158) to read:

"to account for the limitation that in shallow sequencing the expected coverage is $\leq 1x$ for the vast majority of detected sites. "

- The authors describe the generation synthetic nanopore "reads". In a real nanopore run, coverage probability is non-uniformly distributed, as certain genomic regions are less accessible, and in case of cancer, chromosomal regions may be gained/amplified or present as monozygous state or even deleted and this will have strong impact on the nanopore read distribution. Please comment how this was addressed and by the authors an how and how non-uniform coverage might affect the model's performance.

The reviewer is correct, and we have clarified this in the manuscript. The models are trained with uniform, but very sparse coverage. Because of this, the model cannot learn anything from coverage biases. In practice the models are applied on very sparse, nanopore sequence reads, which may be enriched or depleted for specific regions. However due to the sparsity, only very few regions are covered. Those sites that are sequenced may be covered due to a bias (ie. amplification), but the model only "sees" the methylation state, which is also binarized if >1X coverage is obtained. The model can thus not infer anything from the

coverage bias, which is perhaps for the best as the training data only contains 2801 samples, and thus only 2801 different CNV profiles and if the model would be able to learn these, it could lead to overfitting.

We tested the robustness of our classifier to samples that exhibit non-uniform CNV profiles. For this, we changed our simulation to take into account specific CNV profiles, by sampling sequencing reads according to a probability dependent on the genomic position. We then compared the results of our 'normal' cross-validation simulation, with the non-uniformly simulated 40 minute sequencing runs. Four different non-uniform profiles were simulated:

- Two clinically relevant profiles: which we call glioblastoma (Chr7 amplification and Chr10 deletion) and oligodendroglioma (Chr1 and Chr19 deletions) due to their prevalence on these CNS types.
- Two extreme cases to test for robustness: which we call odd (all even chromosomes deleted) and even (all odd chromosomes deleted).

Our results show that the model is extremely robust. The observed changes in F1-scores are minute for any of the non-uniform CNV profiles compared to the uniform CNV profile simulations that were used for training. Together with all the nanopore sequencing runs on real samples (which also exhibit non-uniform CNV profiles), this clearly demonstrates that Sturgeon is very robust to the sampling due to copy-number changes.

CNV simulation analysis. Sturgeon performance (F1-score) simulations test fold samples from the reference dataset from Capper et al., 2018. We compared uniform genomic sampling ($N=5,000,000$) against: extreme sampling (only odd or even chromosomes) and biologically relevant CNV profiles for glioblastoma (Chr7 amplification and Chr10 deletion) and oligodendroglioma (Chr1 and Chr19 deletions), ($N=500,000$ for each CNV profile). We evaluated all four Sturgeon submodels on the test fold samples, each sample was simulated 50 times using test fold seeds.

- The use of Illumina DNA methylation array data as a proxy for simulated nanopore reads is an important and innovative solution. A figure quantifying the agreement (or discrepancy) between these two technologies would be helpful in validating this step.

This is a good suggestion. We have now included a more extensive analysis of methylation arrays versus nanopore sequencing. We show that the major source of discrepancy is,

perhaps as expected, binarization of heterogeneously methylated sites. However, dropping these from analysis will negatively influence the number of usable sites, which is why we still include them.

Changes to manuscript:

-We added supplementary figure 1 and supplementary table 1 to elaborate on the differences between Infinium methylation calling and nanopore sequencing, and changed the text (lines 162-167):

“Random noise, to account for our expectation that nanopore methylation aware sequencing has an expected discrepancy rate of ~10-15% when compared to binarized EPIC arrays, resulting from a combination of heterogeneous methylation states across alleles and cells and methylation calling errors “

- In Section 2.4 the manuscript states "To assess the performance of Sturgeon in a realistic setting...". To me the setting that follows is not a realistic intrasurgical setting but merely an analysis of retrospective frozen samples. This should be rephrased.

We have rephrased this sentence accordingly (line 303).

“Next, we assessed the performance of Sturgeon on real nanopore sequencing data.”

- The frozen test samples only cover relatively few of the classes that are available in the classifier. This should be expanded as it is not clear whether performance generalizes to all classes. The authors should also consider to add adult cases.

We thank the reviewer for this comment. For frozen samples we are limited to samples obtained from clinical practice in the past few years, and some entities are simply very uncommon. We do however include >400 cases (including both adults and children) from a publicly available dataset which cover a more broad range of classes (Figure 3).

We agree with the reviewer that the Sturgeon classifier is not limited to pediatric cases. Therefore, we have now also applied Sturgeon intraoperatively in 5 adult glioma cases. Together the majority of classes are now sequenced at least once.

Changes to manuscript:

- We added 21 new intraoperative samples, including 5 adult gliomas.
- We changed the order and emphasis to focus more on nanopore sequencing samples.

- Sturgeon is intended to be applied in an intraoperative setting. In such a setting tumor cell content cannot be controlled as well as in formalin fixed paraffin embedded samples.

Indeed, two of the four samples with true intraoperative analysis had technical difficulties with no or few tumor cells. Therefore, it seems highly relevant to test the robustness, validity and reliability of Sturgeon in a larger cohort of intraoperative samples including low tumor cell samples. This should be done by 1) analyzing a far larger number of true intraoperative samples to observe how often low tumor content phenomenon arises and how Sturgeon classifies such samples 2) analyzing a number of cases with histologically proven low tumor cell content and determining the performance of Sturgeon for those. In addition, the methodology employed in the manuscript for generating synthetic nanopore reads could be extended to create *in silico* mixtures of tumors with precisely defined ratios, in order to probe the sensitivity of the model to this phenomenon. Single-read-level analysis might also offer a solution to cell-type mixture deconvolution in realistic data.

We thank the reviewer for this insightful comment. We agree that a larger set of samples and a higher diversity in sample classes are important factors to gauge Sturgeon performance. Therefore, we have applied Sturgeon intraoperatively in an additional 21 samples. We now also include 5 adult cases, which were moreover processed and analyzed in a different center (Amsterdam University Medical Center), further demonstrating the robustness of our approach.

Furthermore, we acknowledge that intraoperative sequencing faces several practical challenges, which is to obtain high tumor purity samples during surgery. We have therefore experimented with a streamlined workflow, where we now take several samples which are all split in two. One of two parts is used for DNA isolation, while at the same time the other part of the samples are analysed by the pathologist, who then calls to the genomics lab to instruct which of the samples has the highest tumor cell percentage and should be sequenced.

We also simulated a range of tumor purities to estimate the minimum tumor cell percentage that is viable for analysis. We note that the Capper reference dataset, from which we simulate our training data, is not 100% pure, and ranges between 42-87% with a median 66% purity. We simulate impure samples by mixing a fraction of reads from a tumor and a control sample. We recalculate the simulated purity as the combination of the original sample purity and the added fraction of control sample (e.g. 50% purity reported by Capper and 50% added control sample = 25% simulated sample purity). Our simulation results show that our model can comfortably handle >50% pure samples, below that percentage, the model starts shifting towards less confident predictions first, and then to predicting the sample as a control. Reassuringly, the model does not confidently (>0.95 score) predict other tumor classes, and rather produces lower confidence scores which most of the time fall into the control class or the correct class.

Purity plot (part of figure 3): Each panel represents increasing simulated Nanopore sequencing. The x-axis indicates the simulated sample purity, the y-axis indicates the fraction of simulations that fall into each category. Categories: confidently correct tumor class (dark green), correct tumor class (light green), confident control class (dark yellow), control class (light yellow), incorrect tumor class (grey), confident incorrect tumor class (black). (This plot shows >20 million simulations).

Changes to manuscript:

- We added a section on sample purity (and simulations to seek the effect of sample impurity), in lines 269-300 and included in figure 2.
- We include a list of estimated tumor fractions from previously conducted intraoperative histology cases in supplementary table 4.
- We added 21 new intraoperative samples.
- We added suggestions to prevent sequencing samples with low purity

• On page 12, the manuscript states: "We find that Sturgeon outperforms nanoDx, the patient-specific random forest classifier, as it is able to correctly predict 9 additional samples (Supplementary figure 12)"; However, from looking at supplementary Figure 12, it is not clear how this conclusion should be reached --i.e. which 9 additional samples are being referred to, and how Sturgeon outperforms nanoDx. Supp. Fig. 12 seems to show 27 cases where nanoDx is correct where Sturgeon is incorrect, compared to only 26 cases vice-versa (i.e. nanoDx slightly outperforming Sturgeon). Please clarify.

We apologise for the confusion in the plot. We have expanded the figure description to ensure that it is easier to understand. This UpSet plot (now moved to Supplemental figure 17b) shows:

- Horizontal bars show the number of samples for each case Sturgeon/nanoDx and correct/incorrect. So Sturgeon correctly classified 383 samples and nanoDx correctly classified 374.
- Vertical bars show the overlap between the different sets, marked with the black circles on the grid. Therefore the vertical bars indicate that:
 - 354 samples were correctly predicted both by Sturgeon and nanoDx.
 - 29 samples were correctly predicted by Sturgeon, but incorrectly by nanoDx.

- 20 samples were correctly predicted by nanoDx, but incorrectly by Sturgeon.
- 12 samples were incorrectly predicted by both methods.

Minor points

- Font labels for panels ("a", "b", "c") are inconsistent between figures.

We thank the reviewer for their remark and adjusted the font labels.

- Fig 2d It is unclear what the "*" symbol signifies.

This panel is now moved to supplementary figures and properly annotated.

- Fig. 3 might be condensed, or moved to a supplement.

In light of this and other comments, we reworked this figure and other figures to condense the simulation results and focus more on nanopore sequencing results, and to also make room for the simulated performance at increasing levels of sample impurity.

- The caption in Figure 4 states "Asterisks indicate the first time point where the score of the correct class was higher than 0.8." The asterisks in the figure actually appear to correspond to the point where the threshold 0.95 is crossed, not 0.8 as stated in the caption.

We thank the reviewer for bringing this mistake to our attention, and it is now corrected.

- In Fig 4a, 5a - the x-axis marks are overly dense and non-uniform, making it difficult to draw comparison between adjacent plots. Fewer, evenly spaced ticks at shared positions (e.g. 5000, 10,000, 15,000) would ease readability.

We have replaced the x-axis annotation with the pseudotime, so it is therefore now directly comparable between adjacent plots.

- In Fig 4a,
 - the abbreviations may not be known to the average reader.
 - A few illustrative examples should be employed, with the remainder moved to the supplement.

We have moved most of the individual sample plots to the supplement. We have now, on the main figure:

- 2 example plots of individual samples.

- A single plot with the correct class classification for all the samples.

- In Fig 4c

- different symbols (diamonds/squares/shapes, etc.) should be used instead relying only on colours, as the orange and red are difficult to distinguish for colour-blind readers
- The legend should clarify that colour codes denote when "only NanoDx" is "correct", likewise "only Sturgeon";

We have moved Figure 4c, which compares Sturgeon to nanoDx to Supplementary Figure 17a. This figure has been replaced with the same figure but now class colors are used instead to just showcase the results of Sturgeon. Correct and incorrect classifications are now differentiated by shape and edge color. Incorrect classifications are on top of correct classifications. We also include numbers indicating the number of correct and incorrect classifications at different thresholds.

- On Page 12: "For each sample we generated 200.000 reads (3.9Gb)" the abbreviation for gigabases should be stated explicitly, to avoid confusion with "Gigabytes" (of data)

We have adjusted this accordingly, also on other occasions.

- In Fig. 5a, the caption states "only correct classification classes are displayed", with upward-sloping circle data points corresponding to the two models; it is unclear what the downward-sloping diamond symbols represent. Please recheck this figure.

We have reworked this figure, similarly to the comments provided for Fig4a. We now show a single example brainstem sample, with the results from both the general and the brainstem-specific classifier. And added a figure where it shows the robustness results of all the samples. Individual results for all the samples are moved to the supplement.

- Figure 19 from the supplement is missing. Only 18 figures are shown.

We have reworked the supplementary figures and all referrals should now be correct.

- The supplementary video on YouTube has no audio; perhaps it was redacted for the privacy of laboratory staff, but a note of clarification to that effect should be added.

We indeed removed the audio as it is not informative and also not in English. We now also clarify this in the main text.

- Word economy: multiple long sentences are occasionally used where fewer, shorter sentences could convey the same information more clearly. The abstract in particular could be condensed.

We have had a critical look at the text and made several adjustments, please refer to the “tracked changes” document for details. We have reworked the abstract to make it more condensed, which is now <200 words as recommended by the journal’s guidelines.

- It is appropriate that the bulk of the paper is written to ease reading, while more detailed discussion is relegated to, e.g. methods. However, for a more full picture, references to detailed sections should be provided where an overview leaves room for uncertainty. More sentences like "as described in section x/y ..." Are necessary. For example:
 - page 12- subsampling methodology, ( "refer to methods"....)
 - The definition of "correct" vs. "incorrect" classifier predictions require clearly defined ground truth not established until section 4.9.
 - Page 3: "Our final models are trained on 14 million and validated on 4 million simulated nanopore runs," It is not yet clear at this point what a "run" represents; the reader should be referred to sec. 4.7 (page 23) where this is clarified.

We have specified the specific methods sections when we refer to them.

- On Page 13: it is written "We note that practically all of the misdiagnoses arise from two of the 35 samples". This qualitative statement should be made quantitative : e.g. "We note that x of y misdiagnoses..."

This section is now removed to put more emphasis on intraoperative cases, and we refrain from making similar qualitative assessments.

- The F1-score metric should be defined when introduced for the benefit of general readers.

We agree that the metric should be explicitly defined. We have added a definition of the F1-score metric in the Model evaluation section of the methods.

- On Page 4 the manuscript states: "Sturgeon therefore employs a data augmentation approach to effectively upsample the number of training samples available. This approach also allows for class-balancing by upsampling small classes relatively more compared to larger classes."
 - Here, the 2nd usage of "upsampling" is a clear reference to balancing rare-vs-common class types. However, the first usage of "upsampling" is unclear from the context and should be clarified.

We have rephrased this section line (143-149):

“Sturgeon therefore employs a data augmentation approach to upsample the number of training samples available: thousands of unique shallow nanopore sequencing experiments are simulated from each methylation profile. The number of sequencing experiments simulated from each profile depends on the number of profiles in the same class. Class balance is achieved by simulating more sequencing experiments from profiles in smaller classes.”

Reviewer Reports on the First Revision:

Referees' comments:

Referee #2 (Remarks to the Author):

The authors have addressed all my previous comments to the best of their ability. In particular, they attempted to re-sequenced five samples with adaptive sampling enabled on half of the channels. They encountered some technical difficulty (related to GPU utilization on their workstation), so in the end only succeeded in running it on a GridION device.

They were unable to incorporate CNV profiles in the Machine Learning model. While it would have been ideal to enable methylation plus CNV for tumor subclassification, due to practical reasons, I think the explanations from the author are acceptable.

For my comments on neural network, they have expanded the "Neural network architecture" section to elaborate on the machine learning details.

"(Supplementary figure x, copied below as well)." I think they meant supplementary figure 1 as I saw this in the supplementary materials.

The authors emphasized tumor heterogeneity in their response. I think it is an important point to make in the main manuscript as well: they are different from cancer cell lines where a single cutoff of beta value can easily binarize methylated vs non-methylated sites. I wonder for these samples that they tested, if there is a good way to show subclonal heterogeneity and how it influences the results of shallow Nanopore sequencing. Even measures like stromal contamination from alternative approaches would be helpful for readers to understand the limitations of the study.

A minor comment on Figure 4: I am not sure if the current form is appropriate to show in main text, especially given the relatively low quality (informativeness) of the several photos. I would think simplified illustrations are better used here to show readers of the procedures.

Referee #3 (Remarks to the Author):

We thank the authors for their resubmission and offer the following remarks:

=== Major points: ===

* Clarification on the architecture of the neural network that was used satisfies our request for clarification. Furthermore, we thank the authors for making their code publicly available online, and look forward to using this resource.

* As the authors note, the issue of coverage bias due to chromosomal abnormalities is unavoidable,

but we are persuaded by the tests performed by the authors showing that the methodology is robust to this confounding effect.

* Regarding sample purity, we thank the authors for implementing our suggested tests using *in silico* mixtures. We are satisfied that the limits of the methodology with respect to sample impurity have been adequately probed.

=== Minor issues still outstanding: ===

* Minor misspellings (such as, e.g., "substampling") should be corrected with a spell-checking tool.

* Regarding Supplementary Figure 17a.: The rebuttal document states "Correct and incorrect classifications are now differentiated by shape and edge color."

The shapes shown in supplementary figure 17a are in fact still all circular. Moreover, the shades of Red and orange chosen for two of the classes are particularly problematic for readers with a type of colour-blindness. Again, distinction based on shapes (--e.g. triangles/squares, which are visible to all readers) is recommended.

Author Rebuttals to First Revision:

Referees' comments:

Referee #2 (Remarks to the Author):

The authors have addressed all my previous comments to the best of their ability. In particular, they attempted to re-sequenced five samples with adaptive sampling enabled on half of the channels. They encountered some technical difficulty (related to GPU utilization on their workstation), so in the end only succeeded in running it on a GridION device.

They were unable to incorporate CNV profiles in the Machine Learning model. While it would have been ideal to enable methylation plus CNV for tumor subclassification, due to practical reasons, I think the explanations from the author are acceptable.

We thank the reviewer for the continued efforts and great suggestions. Without doubt, these have improved the article. For the sake of brevity, we have decided to shorten the description of the copy number variation analysis in the main text and moved the details into an extended data panel.

For my comments on neural network, they have expanded the “Neural network architecture” section to elaborate on the machine learning details.

“(Supplementary figure x, copied below as well).” I think they meant supplementary figure 1 as I saw this in the supplementary materials.

The authors emphasized tumor heterogeneity in their response. I think it is an important point to make in the main manuscript as well: they are different from cancer cell lines where a

single cutoff of beta value can easily binarize methylated vs non-methylated sites. I wonder for these samples that they tested, if there is a good way to show subclonal heterogeneity and how it influences the results of shallow Nanopore sequencing. Even measures like stromal contamination from alternative approaches would be helpful for readers to understand the limitations of the study.

We thank the reviewer for the suggestion, this point broaches two different topics: tumor purity and tumor heterogeneity.

For tumor purity (i.e. collection of samples that contain a significant fraction of stromal cells), we attempted to address the challenge this can pose to correct classification in figure 2, and we acknowledge that the technology is currently limited to relatively pure samples. We expect that future endeavors may allow classification of low purity samples, for example by incorporating signal deconvolution strategies.

As for tumor heterogeneity (ie. different tumor populations within the same sample) we unfortunately have no information regarding this phenomenon in the reference data (thus a heterogeneous methylation state in the reference data may be the result of heterogeneity, but also allelic differences within the same population). Notably, as the training samples contain this heterogeneity, we can assume this is inherently incorporated into the model. However, in heterogeneous tumors, shallow nanopore sequencing will randomly sample reads from the different populations, and we see no obvious way of deconvoluting these populations with shallow data. It should be noted that this does not seem to be a major problem, as in the vast majority of comparisons with post-operative microarray-based diagnosis, the nanopore-based diagnosis is in agreement. With deeper sequencing one could perhaps use a combination of genetic haplotyping and imprinted methylation patterns to discern different populations, however this is beyond the scope of the present work.

A minor comment on Figure 4: I am not sure if the current form is appropriate to show in main text, especially given the relatively low quality (informativeness) of the several photos. I would think simplified illustrations are better used here to show readers of the procedures.

We thank the reviewer for this suggestion, and also because the article needs to be shortened somewhat we have exchanged the pictures for icons.

Referee #3 (Remarks to the Author):

We thank the authors for their resubmission and offer the following remarks:

=== Major points: ===

* Clarification on the architecture of the neural network that was used satisfies our request for clarification. Furthermore, we thank the authors for making their code publicly available online, and look forward to using this resource.

* As the authors note, the issue of coverage bias due to chromosomal abnormalities is unavoidable, but we are persuaded by the tests performed by the authors showing that the methodology is robust to this confounding effect.

* Regarding sample purity, we thank the authors for implementing our suggested tests using in silico mixtures. We are satisfied that the limits of the methodology with respect to sample impurity have been adequately probed.

We sincerely thank the reviewer for taking the time to review our work, and for helping us improve the article.

=== Minor issues still outstanding: ===

* Minor misspellings (such as, e.g., "substampling") should be corrected with a spell-checking tool.

We thank the reviewer for their attentive reading and will correct the spelling errors.

* Regarding Supplementary Figure 17a.: The rebuttal document states "Correct and incorrect classifications are now differentiated by shape and edge color."

The shapes shown in supplementary figure 17a are in fact still all circular. Moreover, the shades of Red and orange chosen for two of the classes are particularly problematic for readers with a type of colour-blindness. Again, distinction based on shapes (--e.g. triangles/squares, which are visible to all readers) is recommended.

We thank the reviewer for this suggestion and have adjusted supplementary figure 17 accordingly by adding shape-based distinctions.